# Single-cell profiling guided combinatorial immunotherapy for fast-evolving CDK4/6 inhibitor-resistant HER2-positive breast cancer

Qingfei Wang[1,2], Ian H. Guldner[1,2], Samantha M. Golomb[1,2], Longhua Sun[1,2], Jack A. Harris[1,2], Xin Lu[1,2,3] & Siyuan Zhang [1,2,3]

Acquired resistance to targeted cancer therapy is a significant clinical challenge. In parallel with clinical trials combining CDK4/6 inhibitors to treat HER2+ breast cancer, we sought to prospectively model tumor evolution in response to this regimen in vivo and identify a clinically actionable strategy to combat drug resistance. Despite a promising initial response, acquired resistance emerges rapidly to the combination of anti-HER2/neu antibody and CDK4/6 inhibitor Palbociclib. Using high-throughput single-cell profiling over the course of treatments, we reveal a distinct immunosuppressive immature myeloid cell (IMC) population to infiltrate the resistant tumors. Guided by single-cell transcriptome analysis, we demonstrate that combination of IMC-targeting tyrosine kinase inhibitor cabozantinib and immune checkpoint blockade enhances anti-tumor immunity, and overcomes the resistance. Furthermore, sequential combinatorial immunotherapy enables a sustained control of the fast-evolving CDK4/6 inhibitor-resistant tumors. Our study demonstrates a translational framework for treating rapidly evolving tumors through preclinical modeling and single-cell analyses.

[1] Department of Biological Sciences, College of Science, University of Notre Dame, Notre Dame, IN 46556, USA. [2] Mike and Josie Harper Cancer Research Institute, University of Notre Dame, South Bend, IN 46617, USA. [3] The Indiana University Melvin and Bren Simon Cancer Center, Indianapolis, IN 46202, USA. Correspondence and requests for materials should be addressed to Q.W. (email: qwang9@nd.edu) or to S.Z. (email: szhang8@nd.edu)

Precision medicine aims to design personalized treatment strategies by taking considerations of the heterogeneity of disease[1,2]. Targeted cancer therapy exemplifies this concept and has become one of the major pillars of modern cancer treatment[3–6]. However, cancer is a consistently evolving multi-cellular ecosystem[7]. Despite the initial clinical response, drug-resistant tumors often emerge after prolonged treatments, which imposes a clinical challenge[8,9]. Significant heterogeneity of the tumor ecosystem, at both genetic and phenotypical level, is one of the primary culprits responsible for emergence of resistant tumors under the selection pressure of targeted therapy[10]. Previous research efforts on the resistance mechanisms have focused on such heterogeneity of tumor cells, demonstrating that the drug-resistant phenotype is a result of selecting rare tumor cells with either preexisting mutations (de novo) or newly acquired mutations (acquired) that confer resistance to specific targeted therapies[11]. In addition, emerging evidence has revealed that tumor microenvironment (TME) factors, collaboratively contribute to the evolving path of the tumor to seemingly inevitable resistance[12].

Modeling the dynamic nature of evolving drug-resistance while capturing a holistic view of both tumor cells and the TME is essential for a systematic interrogation of resistance mechanisms and designing novel strategies[12–14]. Traditionally, exploring molecular underpinnings of drug-resistance relies on either one-pathway-at-a-time approach using in vitro cell culture model or bulk DNA/RNA sequencing approaches comparing sensitive/responsive and resistant clinical tumor samples[15,16]. However, the in vitro models cannot capture the interplay between evolving tumor cells and their microenvironment, and bulk sequencing has limited resolution in revealing tumor heterogeneity or identifying rare cellular events that confer phenotypical significance to drug-resistance[17]. Recent advances of single-cell analyses are revolutionizing the traditional paradigm of studying drug-resistance by enabling a more holistic interrogation of tumor progression in response to treatments at an unprecedented single-cell resolution[18–20]. Single-cell sequencing approaches have effectively revealed intratumoral subclonal hierarchy at diagnosis[21], Darwinian clonal repopulation[19,22], epigenetic reprogramming associated with resistant tumor cells[23] and dynamic changes of tumor-associated immune landscape[18,24–27]. These pioneering studies start to shed light on future clinical management strategies for patients with relapsed resistant tumors[28].

Trastuzumab/Herceptin™, a humanized monoclonal antibody targeting the extracellular domain of human epidermal growth factor receptor-2 (HER2), is one of the most successful targeted therapies for HER2-overexpressing breast cancer[29]. However, both de novo and acquired resistance have been observed in certain patients[30,31]. Using in vitro trastuzumab-resistant cell line model, preclinical studies have mechanistically defined diverse intracellular signaling events conferring resistance[31]. Recently, Goel et. al. revealed that enhanced cyclin D1-CDK4 dependent proliferation confers trastuzumab-resistance in an inducible-HER2 transgenic mouse model[32]. Targeting cyclin D1-CDK4 acts synergistically with trastuzumab and, more intriguingly, elicits anti-tumor immune response[33,34]. In light of such strong preclinical evidence and together with the recent advance of CDK4/6 inhibitors for estrogen receptor (ER)-positive breast cancer[35,36], new combinatorial regimen of CDK4/6 inhibitors plus trastuzumab is under active clinical investigation[37,38]. Despite the promise of this regimen in treating HER2+ breast cancer, one can envision that the resistance will ultimately emerge. Thus, we reasoned that prospectively modeling the tumor evolution in response to a trastuzumab plus CDK4/6 inhibitor regimen will provide valuable insight to the potential acquired resistance mechanisms. Preclinically, proactively exploring alternative therapeutic strategies that target emerging resistance mechanisms to prevent or inhibit resistance will have a direct translational impact on ongoing trials and improve the therapeutic outcome.

In parallel with current clinical trial scenario, here, we prospectively model in vivo acquired resistance to CDK4/6 inhibitor and trastuzumab treatment using a transgenic mouse model. We find that acquired resistance to anti-HER2/Neu antibody plus Palbociclib combination emerges quickly after initial response. Through high-throughput single-cell profiling of the evolving tumors over the course of treatment, including treatment naive, responsive/residual disease and rapidly relapsed tumors, we reveal a distinct immunosuppressive immature myeloid cell (IMC) population infiltrates in the resistant tumors. Next, guided by single-cell analyses, we evaluate the in vivo efficacy of using combinatorial immunotherapy by concomitantly targeting IMCs and enhancing T-cell activity. Further, our rationally designed sequential combinatorial regimens enable a durable response and sustained control of the emergence of acquired resistance in rapidly evolving HER2-positive breast cancers.

## Results

**Rapid emergence of resistance to anti-Her2 and CDK4/6 therapy.** To address the question whether HER2/neu and CDK4/6 inhibition has a sustainable therapeutic effect in advanced HER2-positive breast cancer, we employed the MMTV-neu202^Mul transgenic mouse bearing late-stage mammary tumor (volume > 500 mm$^3$) and examined their response to a continuous anti-HER2/neu antibody (Ab) plus CDK4/6 inhibitor Palbociclib (Pal) treatment. Two weeks of Ab + Pal treatment produced pronounced effects, leading to tumor regression with an average volume reduction of 52.74% (Fig. 1a) and significant suppression of tumor cell proliferation (Supplementary Fig. 1A). In contrast, control mice exhibited an average of 108.4% increase in tumor size over the same period, and Pal or Ab single treatment only showed a mild to moderate effect. Despite the initial significant efficacy of Ab + Pal combination and extended survival to doubled tumor volume (Supplementary Fig. 1B), shortly after tumor regression (2–4 weeks), all combination-treated tumors rebounded and eventually developed resistance (Fig. 1b, Supplementary Fig. 1C–E).

**Single-cell transcriptome profiling of tumor cells.** To explore the molecular underpinnings of the development of resistance, we performed single-cell RNA sequencing (scRNA-seq) on enriched tumor cells (Fig. 1c). First, we used nonlinear dimensionality reduction (t-distributed stochastic neighbor embedding, t-SNE) analysis to examine global transcriptional features across tumor cells from control (naive to treatment), Ab or Pal alone, Ab + Pal responsive/residual disease (APP) and Ab + Pal resistant (APR) tumors/progressive disease (Fig. 1d). We observed distinct distribution patterns and identified six clusters (Supplementary Fig. 2A, B). Generally, individual cells derived from each treatment tended to cluster together (Fig. 1d and Supplementary Fig. 2A–C). Clusters 3, 2, 5, 6, and 1 were largely representing cells derived from control, Ab only, Pal only, APP, and APR tumors, respectively (Fig. 1d, e). One exception to the seemingly mutually exclusive clustering based on treatment was cluster 4, which was characterized by the high expression of proliferation genes such as Top2a, Cdk1, Mki67, and Cenpa (Supplementary Fig. 2D), suggesting that subpopulation of tumor cells conferred tolerance to treatment or adapted to drug selection. Besides the dominant clustering as cluster 1, APR tumor cells also spread into other clusters, indicating the nature of heterogeneity.

To examine the functional implications of gene signatures unique to each cluster, we performed single-sample gene set

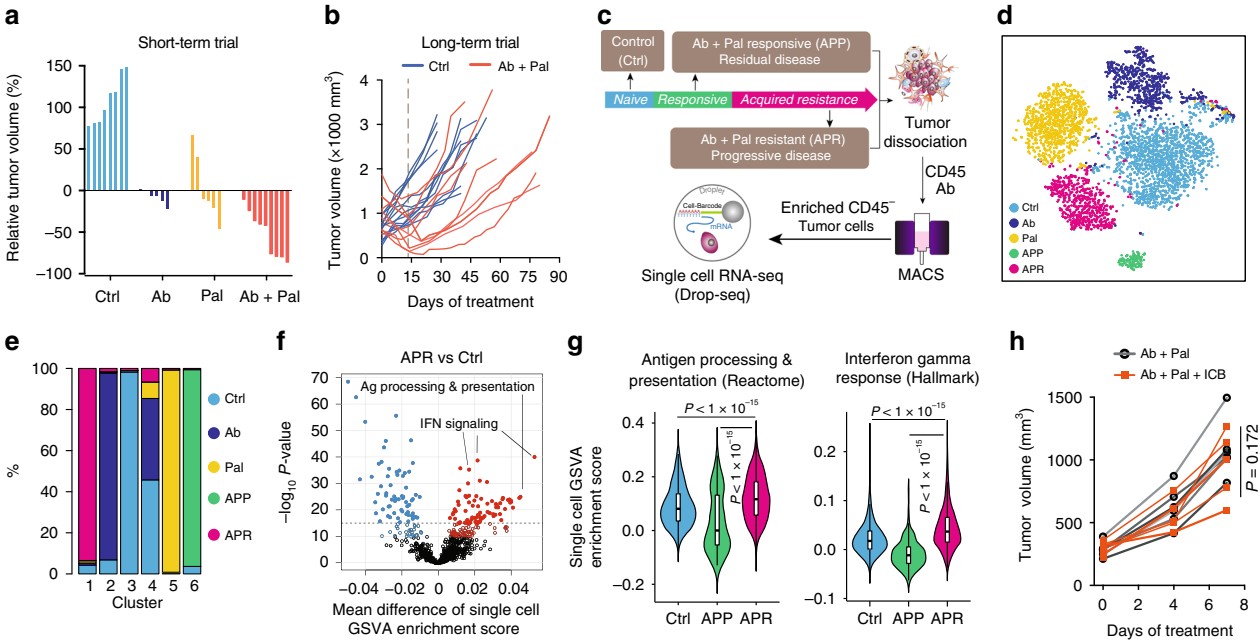

**Fig. 1** Emergence of resistance to Her2 and CDK4/6 targeted therapy and tumor cell scRNA-seq. **a** Waterfall plots showing percent change of tumor volume with 14-days treatment in MMTV-neu mice ($n = 8$, 6, 6, and 9 for Ctrl, Ab, Pal, and Ab + Pal). Ctrl, vehicle treated; Ab, anti-HER2/Neu antibody; Pal, CDK4/6 inhibitor Palbociclib. **b** Tumor volume curves showing tumors rebounded during sustained Ab + Pal combination treatment ($n = 12$ for Ctrl and $n = 10$ for Ab + Pal). **c** Schematic for sample processing, enrichment of tumor cells and Drop-seq based single-cell RNA sequencing. **d** t-distributed stochastic neighbor embedding (t-SNE) plots colored by treatment groups and clustering of 4711 tumor cells derived from Ctrl, Ab or Pal alone, APP (responsive/residual tumors, 10–14 days with Ab + Pal treatment), and APR (resistant tumors/progressive disease, 45–75 days with Ab + Pal treatment). Each point represents a single cell. **e** Abundance of tumor cells with indicated treatment in each cluster. **f** Volcano plots comparing ssGSEA enrichment score of 1053 canonical pathways/gene sets of the C2 collection of Molecular Signatures Database between APR and Ctrl scRNA-Seq data. Each point represents one pathway/gene set. X-axis, mean difference of single-cell ssGSEA enrichment score; Y-axis, −log10 (P-value by t-test). **g** Enrichment score violin plots for single cells in each group for indicated signatures. **h** Volumes of Ab + Pal resistant tumors with treatment combining immune checkpoint blockades ($n = 7$ for Ab + Pal and $n = 6$ for Ab + Pal + ICB). ICB, anti-CTLA4 and anti-PD-1 antibody cocktail. P values by two-tailed Student's t test

enrichment analysis (ssGSEA) focusing on control, Ab + Pal responsive and resistant tumors (Fig. 1f, Supplementary Fig. 2E). Targeting cell-cycle machinery is recognized to be the primary mechanism of action of CDK4/6 inhibitors. GSEA analysis revealed that, overall, G−S-phase cell-cycle transition and mitotic activity were downregulated in APP tumors compared with control tumors, while APR tumors showed a reprogramed cell-cycle machinery with slight enhanced mitotic activity (Supplementary Fig. 2F), which was consistent with Ki67 staining result (Supplementary Fig. 1A, E). APP tumors showed enrichment of genes involved in both death receptor 'P75 NTR signaling' and 'NFκB is activated and signals survival' (Supplementary Fig. 2E, G), suggesting that Ab + Pal treatment induced death signaling and reprogrammed survival signaling to adapt to the treatment. Notably, 'antigen processing and presentation' and 'interferon signaling signatures' were among the most strikingly differential enriched signatures in the APR tumors compared with control and APP tumors (Fig. 1f, g, Supplementary Fig. 2E–H). These results at the single-cell transcriptome level indicated that CDK4/6 inhibitor treatment elicits antigen presentation and stimulate interferon signaling, supporting and extending previous observations[33]. Given that increased antigen presentation and interferon signaling, which suggested an elevated tumor immunogenicity in APR tumors, we next sought to combine immune checkpoint blockades (ICB, anti-CTLA4, and anti-PD-1 antibodies) to overcome or prevent the resistance to Ab + Pal treatment. However, the addition of ICB to the rebound APR tumors showed only modest effect (Fig. 1h, Ab + Pal + ICB), suggesting other factors rather than CTLA4 and PD-1/L1 axis might be the major mediator for the resistance.

**Enrichment of IMCs in resistant tumors revealed by scRNA-seq**. We next investigated the TME factors that could potentially mediate the development of resistance. The observation that more CD45$^+$ leukocytes in both APP and APR tumors compared with Ctrl (Supplementary Fig. 3) led us to focus on the immune compartment. CD45$^+$ tumor-infiltrated leukocytes (TILs) were isolated then scRNA-seq was performed (Fig. 2a). tSNE clustering identified nine clusters among 1444 TILs (Fig. 2b, left). Unlike the distribution pattern of tumor cells which were largely dependent on treatment, a great number of TILs from different groups were mixed together or clustered closely (Supplementary Fig. 4A), suggesting their similar transcriptomic properties. Initial examination of top cluster-specific genes revealed major features of macrophage (e.g., *Apoe*, *Lyz2*, and *C1qc*) in clusters 1 and 2, meanwhile, clusters 8 and 9 showed high expression of NK and/or T-cell genes (e.g., *Nkg7*, *Gramb*, *Cd3g*, *Cd3d*, *Trbc2*, and *Cd8b1*) (Supplementary Fig. 4B). The classification of macrophage, T and NK cells (Fig. 2b, right), was also supported by visualization expression of key marker genes across the single-cell data (Supplementary Fig. 4C). Of note, cells of clusters 4 (327 cells) and 5 (191 cells) displayed high expression of monocyte genes (*Cd14* and *Lcn2*) with the unique expression of *Arg1* and *Xbp1* (Supplementary Fig. 4B–D), which are molecular features associated with myeloid-derived suppressor cells (MDSCs)[39,40]. Cluster 6 (117 cells) showed intermediate expression of cluster 1 and 2-specific genes, as well as cluster 4,5-related genes, suggesting that these cells might be an intermediate state between macrophage and cells of clusters 4 and 5. Therefore, cells of cluster 4, 5, and 6 were annotated as IMCs (Fig. 2b). The above single-cell transcriptome-based profiling and classification of TILs

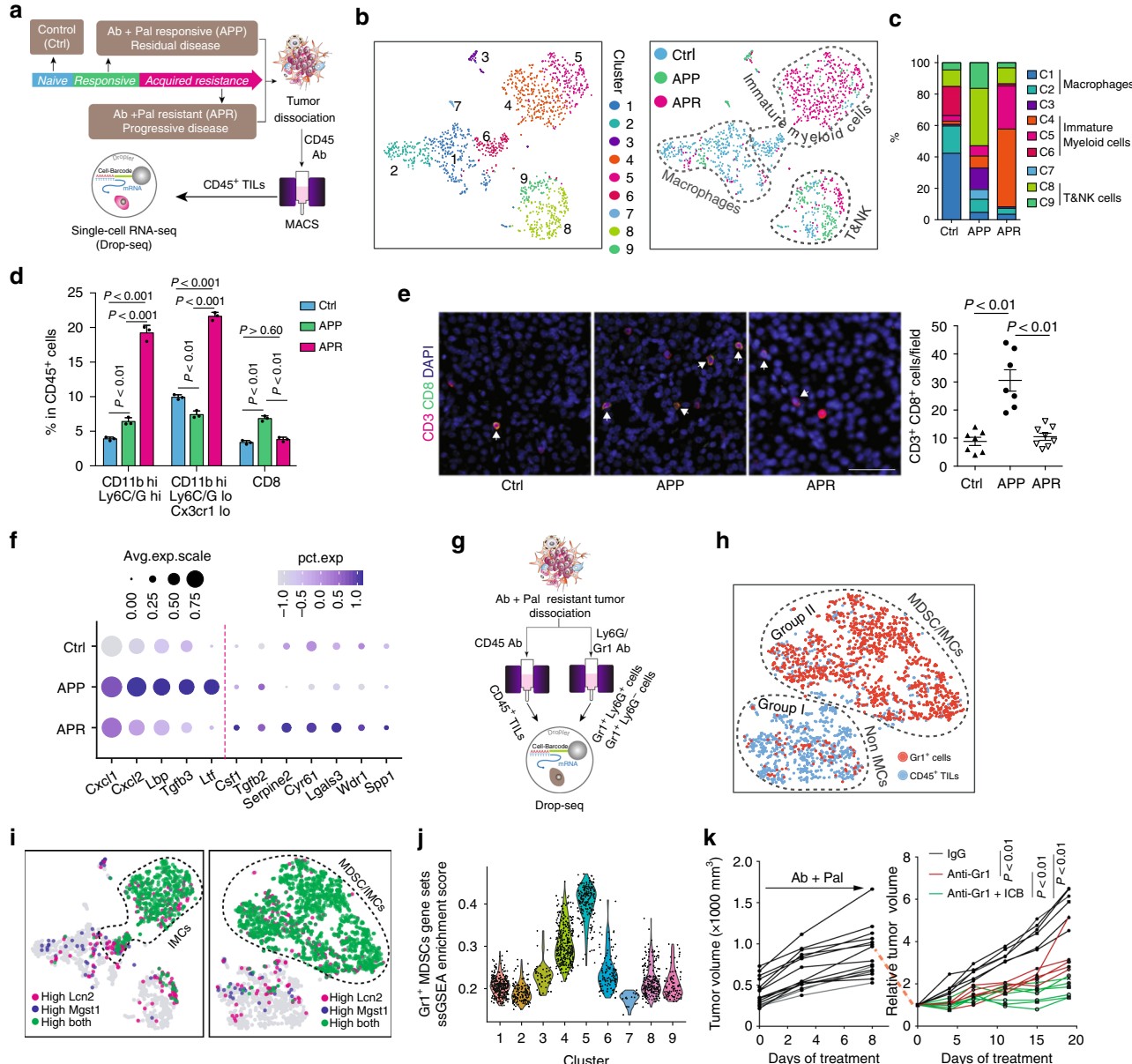

**Fig. 2** Distinct immune milieu in different phenotypes and immature myeloid cells are enriched in resistant tumors. **a** Schematic for tumor dissociation, isolation of tumor infiltrated leukocytes and Drop-seq based scRNA sequencing. Ctrl, vehicle-treated control; Ab, anti-HER2/Neu antibody; Pal, CDK4/6 inhibitor Palbociclib; TILs, tumor infiltrated leukocytes. **b** Clustering of 1444 TILs derived from Ctrl, APP, and APR tumors and t-SNE plot colored by clusters (left) and annotation of TIL-clusters on t-SNE plot colored by treatment groups (right). **c** Abundance of each cluster (as clustered and annotated in **b**) in TILs with indicated treatment. **d** Analysis of immune cell populations among CD45$^+$ TILs by mass cytometry. **e** Representative images and quantification of CD3 and CD8 immunofluorescence staining for Ctrl, APP, and APR tumors. Arrows indicate CD3 and CD8 double-positive cells. Scale bar, 50 μm. **f** Single-cell transcriptional analysis of tumor-produced cytokines and chemokines. The size of each circle reflects the percentage of cells in a treatment group where the gene is detected, and the color intensity reflects the average expression level within each treatment group. **g** Schematic for tumor dissociation, isolation of CD45$^+$ TILs and Gr1$^+$ cells and scRNA-seq. **h** t-SNE plot of scRNA-seq data containing 1153 CD45$^+$ TILs and 1318 Gr1$^+$ cells. **i** Overlay of marker genes for Gr1$^+$ cell population identified from experiment of **g** on t-SNE plots derived from **b** and **g**, respectively. **j** Violin plot showing distribution of ssGSEA enrichment score among TIL-clusters identified from **b**. MDSCs signature generated by experiment as shown in **g** was used for enrichment analysis. **k** Growth of APR tumors by adding Gr1 antibody and ICB. APR tumors were transplanted to recipient MMTV-neu mice and treated with Ab + Pal first to acquire the resistance phenotype (left) and then adding IgG (n = 6), anti-Gr1 (n = 5) or anti-Gr1+ICB (n = 7) treatments were followed (right). Each point represents a single cell in **b**, **h**, **i**, **j**. Error bars represent SD in **d** and SEM in **e**. P-value in **d**, **e**, **k** by one-way ANOVA with Tukey's test

indicated a distinct immune milieu among control, Ab + Pal responsive and resistant tumors. The responsive tumors contained a higher frequency of T and NK cells while the resistant tumors were dominated by IMCs (Fig. 2c).

To connect the canonical cell surface markers with the observed transcriptome heterogeneity of TILs, we profiled the

TILs of control, APP, and APR tumors using CyTOF. CD45$^+$ live cells were analyzed and we observed an increase of CD11b$^{high}$ myeloid cells while a decrease of CD11b$^{low}$ cells in APR tumors, and more T and NK cells in APP tumors (Supplementary Fig. 5A–C), which was consistent with the trend of scRNA-seq profiling and classification. Lymphocyte antigen 6 complex

(Ly6C/G) and chemokine (C-X3-C motif) receptor 1 (Cx3cr1) are valuable markers with both phenotypic and functional significance for myeloid cells. Closer examination of CD11b$^{high}$ myeloid cells showed an increase of CD11b$^{high}$ Ly6C/G$^{high}$ (19.23% in APR tumors compared with 3.91% and 6.41% in control and APP tumors, respectively) and CD11b$^{high}$ Ly6C/G$^{low}$Cx3cr1$^{low}$ (21.63% in APR compared with 9.94 and 7.41% in control and APP tumors, respectively) subpopulations in APR tumors (Fig. 2d, Supplementary Fig. 5D). Of note, CD11b and Ly6C/G (Gr1) are recognized as phenotypic markers of mouse MDSCs. Immunofluorescence staining confirmed a significant decrease of CD8$^+$ T-cell infiltration (Fig. 2e) and increase of MDSCs (Supplementary Fig. 5F) in resistant tumors. Collectively, these observations revealed that APP tumors were infiltrated with more T and NK cells while, in contrast, APR tumors were dominated by IMCs infiltration.

The dominant presence of IMCs suggested an immunosuppressive microenvironment in APR tumors. To understand the possible mechanisms involved in the transition of the immune microenvironment, the effect of Ab + Pal treatment on expression of cytokines and chemokines was investigated, as tumor-produced factors are critical for the recruitment and functional properties of TILs[14]. Single-cell transcriptional analysis of tumor cells revealed that several secreted factors involved in recruitment or chemotaxis of myeloid cells were increased, including *Cxcl1*, *Cxcl2*, *Tgfβ3*, and lactotransferrin (*Ltf*) after short-term Ab + Pal treatment (Fig. 2f). Ltf has been reported as a driver for accumulation and acquisition of immunosuppressive activity of MDSCs[41]. On the other hand, expression of multiple cytokines and chemokines associated with myeloid cell recruitment and differentiation, including *Csf1*, *Tgfβ2*, *Serpine2*, *Cyr61*, and *Lgals3* were upregulated in cells from APR tumors (Fig. 2f). Colony stimulating factor 1 is important for development and activation of MDSCs[39,40]. These data indicate that tumor cells are capable of evolution/adaptation through the production of multiple immunomodulatory factors to establish an immunosuppressive environment to acquire and sustain resistance to Ab + Pal treatment.

**Characterization of scRNA-seq annotated immature myeloid cells.** Noticeable in APR tumors, the IMCs annotated by scRNA-seq (clusters 4 and 5) possessed certain molecular characteristics of MDSCs. This observation led us to explore the potential association between the transcriptome profiling identified IMCs and the surface markers defined MDSCs through transcriptomic analysis. First, we isolated tumor infiltrated Gr1$^+$ cells (including Gr1$^{high}$Ly6G$^+$ and Gr1$^{dim}$Ly6G$^-$ populations) and found that these cells inhibited the proliferation of CD4+ and CD8+ T cells in vitro (Supplementary Fig. 6A), an important functional characteristic of MDSCs[39]. Next, Gr1$^+$ cells and CD45$^+$ TILs were isolated in parallel from the transplanted APR tumors and scRNA-seq was performed (Fig. 2g). Unsupervised clustering separated 2471 cells into two apparent subgroups: group I was predominantly from CD45$^+$ TILs while group II was dominated by Gr1$^{high}$Ly6G$^+$ and Gr1$^{dim}$Ly6G$^-$ cells (Supplementary Fig. 6B). Group I cells showed high expression of macrophage genes and T/NK cell-related genes, while group II cells exhibited enriched expression of MDSC-related genes, *Arg1* and *Xbp1* (Supplementary Fig. 6C). Thus, these two groups of cells were annotated as non-IMCs and MDSC/IMCs, respectively (Fig. 2h). Based on marker genes of group II cells, we generated Gr1$^+$ MDSCs signature (Supplementary Table 1). We found that *Lcn2* and *Mgst1*, two of the marker genes of Gr1$^+$ MDSCs identified by scRNA-seq analysis, were also specifically present in previously identified IMCs-related cells (Fig. 2i). Indeed, flow sorting and qPCR of APR tumors showed significant higher expression levels

of both *Lcn2* and *Mgst1* in Gr1$^+$ cells compared with T cells and macrophages (Supplementary Fig. 6D). Further, GSEA using our custom experimentally generated Gr1$^+$ MDSCs signature revealed that those genes were also enriched in the annotated IMCs, particularly in cluster 5 cells (Fig. 2j). This analysis demonstrated that transcriptomic profiling identified IMCs (predominately presented in the APR tumors) displayed similar transcriptome profiles to previously defined Gr1$^+$ MDSCs. MDSCs have been sub-grouped as Gr1$^{high}$Ly6G$^+$ and Gr1$^{dim}$Ly6G$^-$ MDSC, which largely reflect granulocytic/polymorphonuclear and monocytic lineage of MDSCs[39,40]. Interestingly, based on single-cell transcriptome profiles, in our case, the Gr1$^{high}$Ly6G$^+$ and Gr1$^{dim}$Ly6G$^-$ cells were clustered closely or mixed together (Supplementary Fig. 6B), suggesting a similarity of their transcriptomes, despite their distinct expression of surface markers.

**Depletion of IMCs sensitizes APR tumors to ICB treatment.** We next assessed whether the increased Gr1$^+$ MDSCs were functionally important for Ab+Pal resistance. After confirming the resistant phenotype of transplanted APR tumors (Fig. 2k, left), the mice were further treated with either anti-Gr1 antibody or anti-Gr1 plus ICB. MDSCs depletion with anti-Gr1 antibody inhibited growth of APR tumors (Fig. 2k, right). Notably, addition of ICB showed further tumor inhibition (Fig. 2k, right), indicating that MDSCs were not only involved in promoting APR phenotype but also in hindering maximal efficacy of ICB.

**Identification and selection of cabozantinib (Cabo) to target IMCs.** Motivated by the above results, we sought to modulate or target IMCs in APR tumor to overcome Ab + Pal resistance. With a goal of potentially repurposing existing drugs to combat the resistance, we screened the drug target portfolios of FDA-approved small molecular protein kinase inhibitors (PKIs) against the single-cell transcriptome of TILs. We observed that in addition to EGFR and/or HER2 inhibitors, Cabozantinib target genes (*Met, Kit, Axl, Kdr/Vegfr2,* and *Flt3*) and Lenvatinib target genes (*Vegfr1/2/3, Pdgfr, Fgfr, Kit,* and *Ret*) were significantly enriched in TILs from APR tumors (Supplementary Fig. 7A, B). Cabo, an orally bioavailable tyrosine kinase inhibitor, is approved for metastatic medullary thyroid cancer and renal cell carcinoma. Cabo also showed promising clinical activity for metastatic breast cancer in a phase 2 trial[42] and is being further investigated. This prompted us to conduct an in-depth examination of Cabo. Unlike TILs (Fig. 3a, left), the enrichment of Cabo target genes in APR tumor cells showed no significant difference compared with control and APP (Supplementary Fig. 7C). Interestingly, IMCs showed the highest average enrichment score (Fig. 3a, right) and clusters 4, 5 (IMC clusters) possessed more cells with a relatively high enrichment score (Supplementary Fig. 7D). Specifically, IMC clusters contained a higher percentage of *Kit* and/or *Met* expressing cells compared with either T&NK cells or macrophages (Fig. 3b). Moreover, the IMC population derived from APR tumors were largely composed of *Kit* and/or *Met* expressing IMCs (Fig. 3c). Consistently, Gr1$^+$ MDSC/IMCs isolated from APR tumors also showed much higher percentage of *Kit* and/or *Met* expressing cells than other non-IMCs (including macrophages, NK and T cells) (Supplementary Fig. 7E, F). Indeed, qPCR confirmed higher expression levels of *Kit* and *Met* in CD45$^+$ TILs from APR tumors and Gr1$^+$ MDSCs/IMCs showed the highest expression of *Kit* and *Met* among the sorted cells (Supplementary Fig. 7G, H). Altogether, these single-cell analysis and validation suggested that IMCs in APR tumors might be targetable by Cabo.

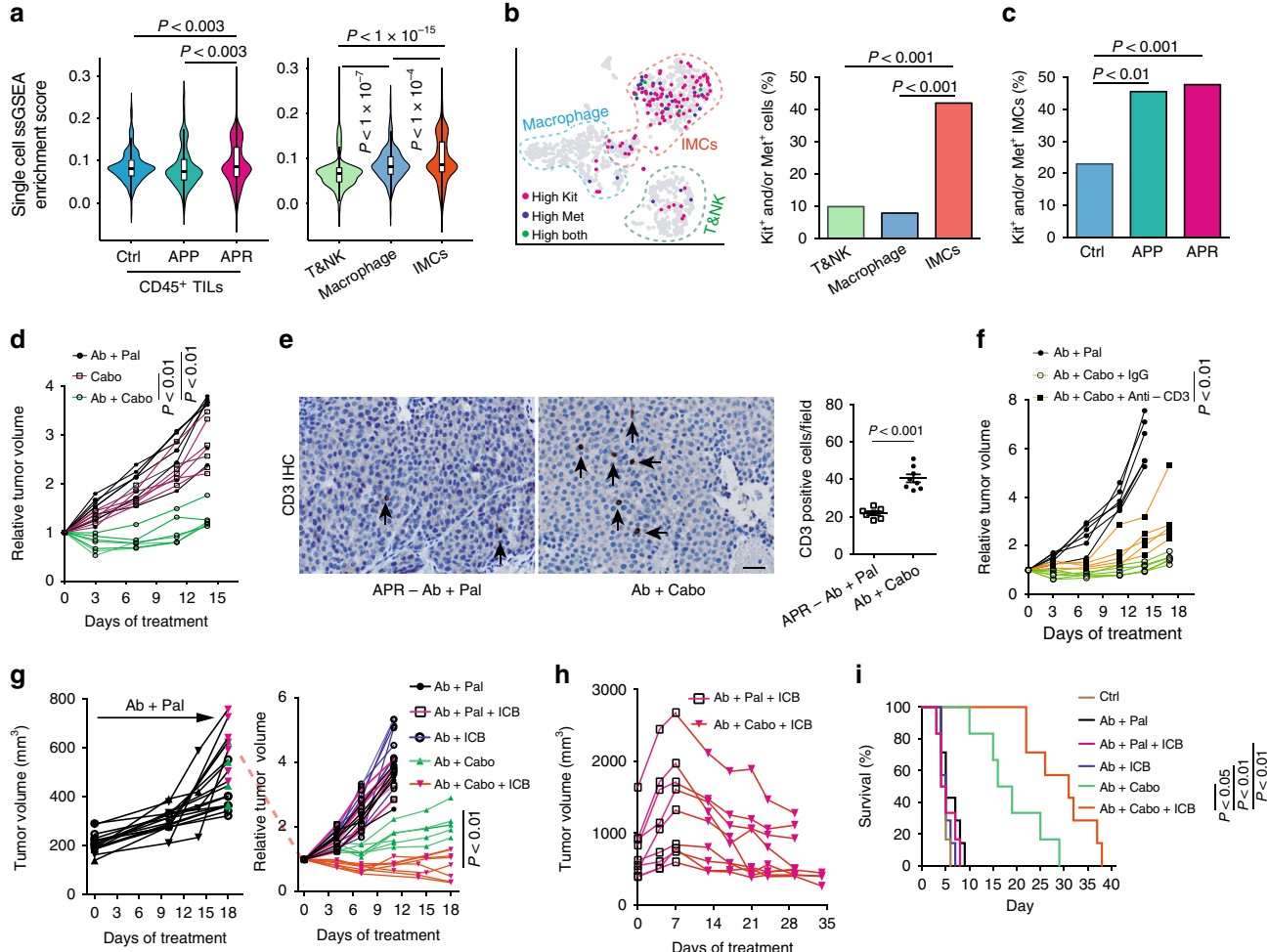

**Fig. 3** Therapeutic strategy for anti-Her2 and Palbociclib resistant tumors informed by scRNA-seq analysis. **a** Enrichment analysis of cabozantinib target genes across single TILs grouped and plotted by different phenotypes (left) or by different immune cell types (right) as annotated in Fig. 2b. **b** Expression distribution of *Kit* and *Met* on t-SNE plot (left) and quantification of *Kit* and/or *Met* expressing cells among different immune cell types (right). **c** Abundance of *Kit* and/or *Met* expressing IMCs among tumors with different phenotypes. **d** Growth of APR tumors with Ab + Cabo treatment (n = 7 for Ab + Pal, n = 6 for Cabo and n = 7 for Ab + Cabo). **e** Representative images and quantification of CD3 immunohistochemistry staining for tumors with Ab + Pal or Ab + Cabo treatment. Scale bar, 20 μm. Error bars represent SEM. **f** T-cell depletion during Ab + Cabo treatment against APR tumors. **g** Relative volumes of APR tumors after treatment with Cabo and ICB. APR tumors were transplanted to recipient MMTV-neu mice and first treated with Ab + Pal to acquire the resistance phenotype (left), then treated with Ab + Pal (n = 6), Ab + Pal + ICB (n = 6), Ab + ICB (n = 8), Ab + Cabo (n = 5), or Ab + Cabo + ICB (n = 7) for 2 weeks (right). **h** Growth of APR tumors after sequential treatment with Ab + Pal + ICB and Ab + Cabo + ICB (n = 9). APR tumors were treated with Ab + Pal + ICB for 1 week then switched to Ab + Cabo + ICB treatment for 3 weeks. **i** Survival time to doubled tumor volume of experiment in **g**. Cabo, protein kinase inhibitor cabozantinib. ICB, immune checkpoint blockades cocktail with anti-CTLA4 and anti-PD-1 mAb. *P*-value by Student's *t* test in **a**, **e**, **f**, **g**, by three-sample Chi-square test in **b** and **c**, by one-way ANOVA with Tukey's test in **d**, by log-rank (Mantel-Cox) test in **i**

## Evaluation of combinatorial strategy informed by scRNA-seq analysis.

To evaluate the effectiveness of Cabo, a potential MDSC/IMCs targeting inhibitor, for treating APR tumors, we again employed the transplantation model similar to previous experiments (Fig. 2k) to establish a cohort of mice with relatively uniform tumors. The transplanted APR tumor bearing mice were either continuously treated with Ab + Pal or with Ab + Cabo. Although Cabo monotherapy at the given dose had no anti-tumor activity, Ab + Cabo treatment significantly inhibited tumor growth (Fig. 3d). Interestingly, Ab + Cabo treated tumors showed increased T-cell infiltration compared with tumors with continuous Ab + Pal treatment (Fig. 3e) and T-cell depletion during Ab + Cabo treatment resulted in significant reduction of tumor suppression (Fig. 3f), suggesting that the optimal therapeutic activity of Ab + Cabo against APR tumors is dependent on T-cell. Next, addition of ICB to Ab + Cabo combination further improved therapeutic efficacy (Fig. 3g). Consistent with the results in Fig. 1h, Ab + Pal + ICB had limited efficacy on APR tumors (Fig. 3g). Histology analysis revealed a significant increase of tissue hypocellularity and a reduced tumor proliferation (Supplementary Fig. 8A, B) in Ab + Cabo and Ab + Cabo + ICB-treated tumors. Furthermore, in another cohort of mice, although adding ICB (Ab + Pal + ICB) had limited effect on APR tumor progression, notably, switching to Ab + Cabo + ICB combination treatment led to tumor shrinkage (Fig. 3h, Supplementary Fig. 8C). Importantly, both Ab + Cabo and Ab + Cabo + ICB treatment greatly extended survival (time to doubled tumor volume) from a median of ~5 days in Ctrl and continuous Ab + Pal treatment group to 17.5 days in Ab + Cabo-treated group and up to 31 days in Ab + Cabo + ICB-treated group (Fig. 3i). Together, these data indicated that Ab + Cabo combination, identified by single-cell transcriptome analysis, was effective in overcoming Ab + Pal resistance, and the addition of immunotherapy using ICB further enhanced the anti-tumor activity.

**Cabo and ICB combination subverts immunosuppressive TME**. It has been previously shown that Cabo could synergize with ICB by attenuating MDSC frequency and immunosuppressive activity in a mouse model of metastatic castration-resistant prostate cancer[43]. Since Cabo alone did not effectively suppressed the APR tumor growth (Fig. 3d), we speculated that the anti-tumor effect of Cabo-containing combinatorial regimen might be due in part to its activity on modulating IMCs in the TME. Next, we performed CyTOF analysis focusing on CD45$^+$ TILs from APR transplants with either continuous Ab + Pal treatment, Ab + Cabo or Ab + Cabo + ICB combination. Both CD11b$^{high}$ Ly6C/G$^{high}$ and CD11b$^{high}$ Ly6C/G$^{low}$Cx3cr1$^{low}$ populations, which were enriched most significantly in the APR tumors (Fig. 2d), were greatly decreased after Ab + Cabo treatment (Fig. 4a, Supplementary Fig. 9). The addition of ICB led to further reduction of the CD11b$^{high}$ Ly6C/G$^{low}$Cx3cr1$^{low}$ population (Fig. 4a). Meanwhile, Ab + Cabo treatment showed a mild or moderate increase of CD11b$^{low}$ populations and NK, CD4$^+$, and CD8$^+$ T cells.

To gain deeper insight into how Cabo and ICB combination modulated the immune response, we next performed scRNA-seq on CD45$^+$ TILs. Examining the expression of immune lineage marker genes among 3168 TILs (Supplementary Fig. 10A, B) identified three major categories: IMCs (clusters 1–3), macrophages (clusters 4–9), and T and NK cells (clusters 10–12) (Fig. 4b). In line with the CyTOF results, this classification of TILs indicated that Ab + Cabo treatment decreased tumor infiltrated IMCs and increased T and NK cells (Fig. 4c). Addition of ICB led to further reduction of IMCs population concomitant with mild increase of T and NK cells (Fig. 4c) and immunofluorescence staining confirmed further increase of CD8$^+$ T-cell (Supplementary Fig. 10C). Meanwhile, enrichment of our experimentally generated Gr1$^+$ MDSCs-related signature was decreased among TILs derived from APR tumors treated with Cabo-containing regimen (Fig. 4d). Particularly, TILs from Ab + Cabo + ICB triple combination exhibited the lowest proportion of cells expressing MDSCs signature (Fig. 4d). Notably, Cabo-containing regimen suppressed proliferation of IMCs as indicated by Ki67 expression in IMCs (Fig. 4e). Moreover, combining Cabo attenuated the enrichment of its target genes among IMCs (Supplementary Fig. 10D) and decreased *Kit* or/and *Met* expression IMCs (Supplementary Fig. 10E) and Met signaling as well (Fig. 4f). In addition, compared with continuous Ab + Pal treatment, Ab + Cabo, or Ab + Cabo + ICB treatment not only promoted infiltration of T and NK cells into the tumors (Fig. 4a-c and Supplementary Fig. 10C) but also enhanced T-cell-related anti-tumor activity, and to a greater extent within tumors after Ab + Cabo + ICB treatment (Fig. 4g). Specifically, pairwise comparison revealed that the enrichment of 'T-cell receptor signaling' and 'Costimulation by the CD28 family' signatures across T&NK cell clusters with Ab + Cabo + ICB treatment were higher than those of with Ab + Cabo treatment (Fig. 4h), indicating enhanced T-cell response by ICB, which subsequently increased therapeutic effect. Immunofluorescence staining showed more Granzyme B$^+$CD8$^+$ T-cell in Ab + Cabo-treated tumors and additional ICB treatment (Ab + Cabo + ICB) resulted in further increase of these cells (Supplementary Fig. 10F).

**Sequential immunotherapy enables a sustained therapeutic efficacy**. Our results have shown that Ab + Pal combination initially inhibited spontaneous late-stage HER2/neu-positive mammary tumor. However, resistance to Ab + Pal treatment emerged in a short period (Fig. 1b). We found increased immunogenicity (with enhanced antigen presentation and interferon signaling) in tumor cells along with distinct immunosuppressive IMCs infiltration in the APR tumors (Fig. 2). Ab + Pal resistance could be effectively overcome by switching to Cabo-containing combinatorial immunotherapy, which reduced IMCs and enhanced anti-tumor immunity. These results prompted us to hypothesize that sequential administration of Ab + Cabo (AbC) or Ab + Cabo + ICB (AbC + ICB) combination after a short period of Ab + Pal (AbP) treatment for anti-tumor immunity priming before the emergence of resistance might achieve a better therapeutic efficacy. To this end, for the control arms, mice bearing spontaneous advanced tumor were continuously treated with either AbP, AbC, or AbC + ICB for 4 weeks, and for the sequential treatment, mice were first treated with AbP for 1 week, then switched to AbC or AbC + ICB treatment for another 3 weeks (Fig. 5a). We observed that sequential regimen with AbC increased progression free survival (PFS) (median of 43 days) compared with AbP continuously treated mice (median of 29 days). Continuous triple combination regimen (AbC + ICB) without the priming stage exhibited comparable PFS (median of 44 days) to that of sequential AbP + AbC treatment. Notably, prior treatment of AbP priming followed by a combinatorial immunotherapy regimen (AbP/AbC + ICB) further increased PFS (median of 53 days) (Fig. 5b). This result suggests that AbP priming is important to recondition the tumor immune microenvironment which makes the tumor more sensitive to AbC + ICB combinatorial immunotherapy.

Clinically undetectable residual tumors might gradually rebound upon discontinuation of the treatment, which imposes a significant clinical challenge. Encouraged by the significant therapeutic efficacy of AbP/AbC + ICB sequential regimen in inhibiting extremely aggressive APR tumors, we next sought to model the clinical scenario of residual disease and test whether the rebounded-tumors would acquire resistance to the sequential combinatorial immunotherapy (Fig. 5c). Strikingly, a second-round of sequential combinatorial immunotherapy was almost as effective as the first round of AbP/AbC + ICB in shrinking the rebounded-tumors (Fig. 5d). Throughout two courses of treatment, sequential combinatorial treatment (AbP/AbC + ICB) was well tolerated without significant weight loss (Fig. 5e). To further explore the sustainability of the sequential regimen in controlling the tumor relapse, we transplanted the residual tumors after the second-round of sequential treatment to a cohort of recipient syngeneic mice. Compared with controls, sequential combinatorial treatment continuously to inhibit tumor progression during the third round of treatment (Fig. 5f), enabling a sustained control of the extremely aggressive tumors.

## Discussion

CDK4/6 inhibitors are one of the most exciting classes of targeted therapies in treating ER-positive breast cancers[35,44]. Recent exciting preclinical studies warrant clinical proposition of CDK4/6 inhibitors to patients with HER2+ breast cancer and multiple clinical trials are currently being conducted[32,37,38]. Through prospective modeling HER2+ breast cancer using the classic MMTV-neu mouse model, we find that long-term efficacy of combined CDK4/6 and HER2-targeted therapy is diminished by acquisition of resistance. By single-cell analyses, we reveal a distinct infiltration of immunosuppressive immature myeloid cells in those resistant tumors and combining Cabo overcomes the resistance and sensitizes them to immune checkpoint blockades. Since there are a number of on-going clinical studies evaluating HER2 and CDK4/6 co-targeting regimen, our study might be valuable in guiding future clinical practice to overcome potential emerging therapeutic resistance.

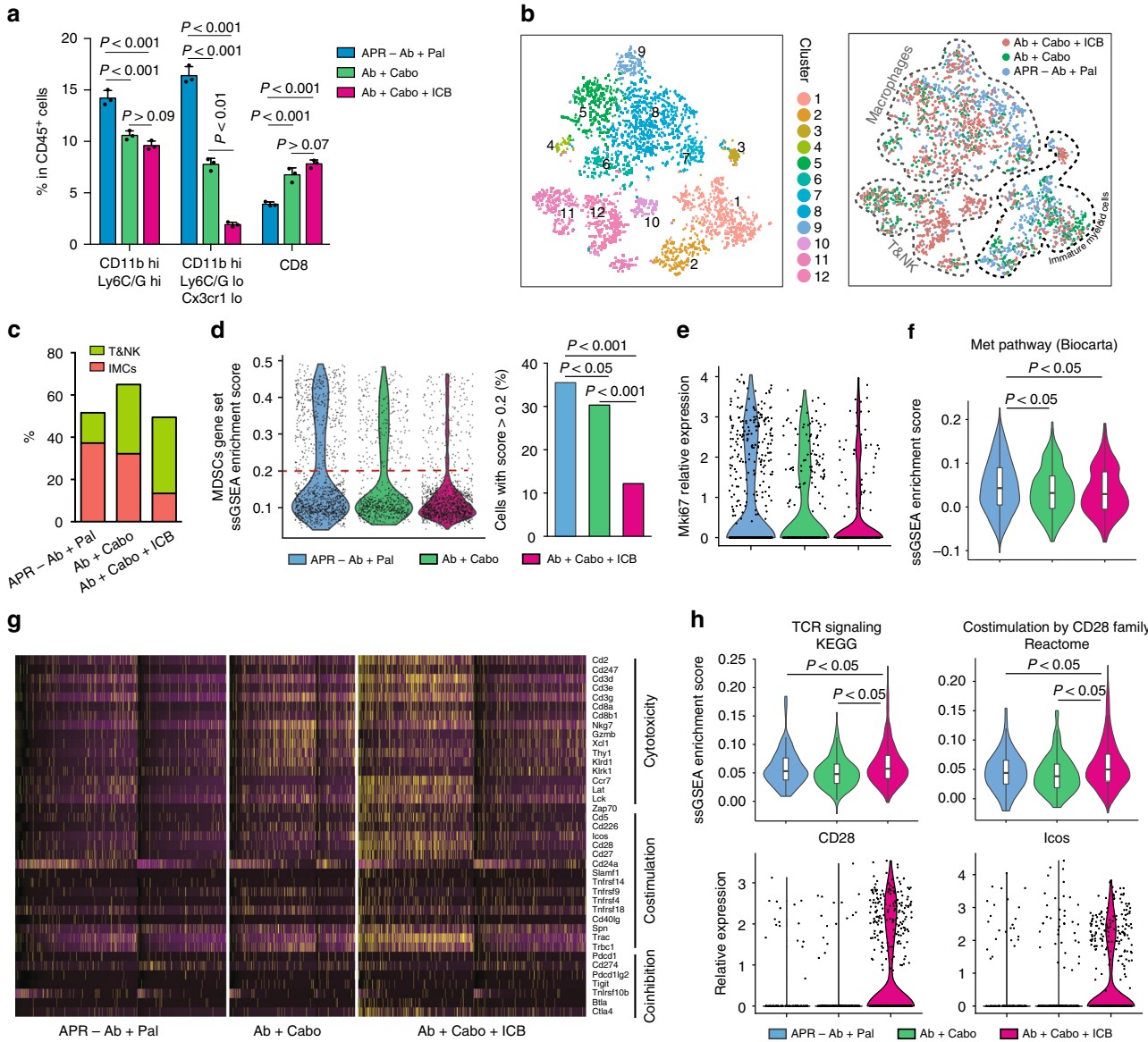

**Fig. 4** Combining Cabo and ICB subverts immunosuppressive TME and enhances anti-tumor immunity. **a** Characterization of tumor infiltrated immune cell populations by CyTOF and their relative abundance among CD45+ TILs after treatment with Cabo and ICB. Error bars represent SD. **b** Clustering and annotation of scRNA-seq data including 3168 TILs derived from APR tumors with continuous Ab + Pal, Ab + Cabo or Ab + Cabo + ICB treatment. Left, t-SNE plot colored by clusters; right, annotation of TIL-clusters on t-SNE plot colored by treatment groups. **c** Abundance of T and NK-cell and IMC clusters (as clustered and annotated in **b**) in the TILs with indicated treatment. **d** Distribution of MDSCs-related signature enrichment score (left) and proportion of cells with high enrichment score in the TILs with indicated treatment (right). MDSCs-related signature was generated from experiment in Fig. 2g (top 300 differential expressed genes of MDSC population) was used for enrichment analysis. **e** Expression of Ki67 in IMC population from tumors with indicated treatment. **f** Enrichment of 'Met pathway' in IMC population from tumors with indicated treatment. **g** The heatmap of T-cell response signature genes across CD45+ TILs. Specific genes from gene sets for T-cell cytotoxicity, costimulation, and coinhibition were shown. Each column represents a single cell. **h** Enrichment scores for 'T-cell receptor signaling' and 'Costimulation by the CD28 family' (upper panel) and expression of CD28 and ICOS (lower panel) in T&NK cell population (as clustered and annotated in **b**). Each point represents a single cell in **b**, **d**, **e**, **h**. *P*-value by one-way ANOVA with Tukey's test in **a**, by three-sample Chi-square test in **d** and by two-tailed Student's *t* test in **f** and **h**

Our study demonstrates that identifying TME changes occurring after treatment is essential for designing more effective combinatorial regimen to combat the tumor evolution. Traditionally, targeted therapy is not effective once the tumor develops new mutation or engages alternative pathways to circumvent the drug target in cancer cells. We reveal a significant increase of immunosuppressive immature myeloid cells infiltration in the APR tumors, which in turn hinders the efficacy of ICB. We believe targeting/modulating these TME changes by defining immune suppressive components and rationally designed

combinatorial regimen will deliver better clinical outcomes, and maximize the utility of ICB in treating breast cancer, especially for APR tumors. In our study, we have demonstrated that sequential administration combinatorial targeted therapy, guided by scRNA-seq, delivers a durable therapeutic efficacy in combination with ICB, leading to a prolonged stable disease of rapidly evolving HER2/neu-positive breast cancer. This long-lasting disease control by such treatment might also engage or enhance immune memory, considering the optimal therapeutic activity of Ab + Cabo with a dependence on T cells. These results provide a

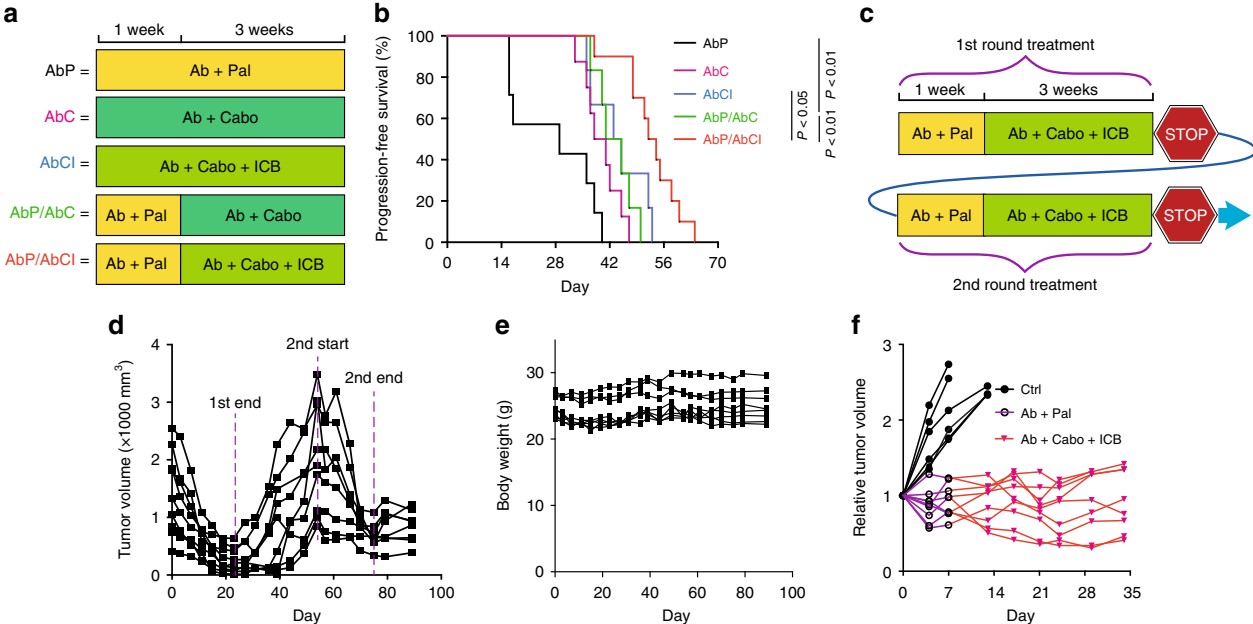

**Fig. 5** Sustained response to sequential combinatorial immunotherapy in rapidly evolving Her2-positive breast cancer. **a** Short-term experimental design for testing efficacy of indicated treatment schedule in MMTV-neu mice with spontaneous, advanced-stage tumors. **b** Kaplan–Meier progression free (without tumor volume increase) survival curves for different treatment and schedule as indicated in **a** ($n = 6$–8). $P$-value by log-rank (Mantel–Cox) test. **c** Long-term experimental design for testing efficacy of sequential combinatorial immunotargeted therapy in MMTV-neu mice with spontaneous, advanced-stage tumors. **d** Tumor volume curves of sequential combinatorial immunotargeted therapy ($n = 9$) as indicated in **c**. **e** Body weight measurements during treatment as performed in **d**. **f** The residual tumors after second-round sequential treatment as performed in **d** were transplanted to recipient syngeneic MMTV-neu mice ($n = 6$ for Ctrl and $n = 9$ for sequential treatment group). Relative tumor volumes of another round of sequential combinatorial immunotargeted therapy were shown

rationale for clinical proposition of combinatorial immunotherapy for HER2+ breast cancer as a strategy to mitigate the emergence of resistance. Our findings also provide insights into how and when to optimally integrate immunotherapies against even aggressive breast cancer with extensive prior treatments.

At the transcriptomal level, the immature myeloid cells identified in our study resemble MDSCs. MDSCs represent a heterogeneous population of largely immature myeloid cells with an immune suppressive activity. Two major subsets (monocytic and polymorphonuclear MDSC) have been identified and characterized[39,40]. However, the current characterization of MDSC relies mostly on the functional level (e.g., ex vivo T-cell suppression assays). The specific and reliable molecular features contributing to the function of MDSCs, especially under the drug-resistance context, have not been well defined at the single-cell level. Defining the mechanistic underpinnings that drive MDSC phenotypes and their immune suppressive properties in tumors is essential for the development of MDSC-specific therapeutic interventions. In this study, we performed scRNA-seq of Gr1^highLy6G+ and Gr1^dimLy6G− cells derived from the APR tumors, which are believed to represent granulocytic/polymorphonuclear and monocytic lineage of MDSCs. Interestingly, to a great extent, they displayed transcriptomic similarity under the APR context. The similarity may reflect common features of these heterogeneous and plastic myeloid subsets, as immunosuppressors in APR tumors[45–47]. Guided by single-cell transcriptome signatures, targeting immature myeloid cells by switching to Cabo, a clinically actionable strategy, suppresses APR tumors and sensitizes them to ICB. Given the abundance of immature myeloid cells in APR tumors and their apparent tumor-promoting functions, targeting or modulating those cells, which have a relatively stable genome compared with cancer cells, is a clinically appealing strategy. We envision that the signature of

MDSCs generated from our study may shed more light on the molecular underpinning of immature myeloid cells and facilitate the development of therapeutic interventions to precisely target MDSCs.

In the present study, although Cabo-containing regimen inhibit immature myeloid cells in the APR tumors, we could not exclude the possible direct effect of Cabo on cancer cells and other stroma compartments. Our findings provided clues as to how immature myeloid cells were dominant in the TME. We found that expression of several cytokines and chemokines involved in recruitment, differentiation and activation of myeloid cells (Csf1/M-CSF, Tgfβ2, Serpine2, Lgals3) were increased in response to Ab + Pal treatment. Studies have shown that tumor cell derived factors can promote bone marrow myeloid progenitor expansion and ultimately increase the number of circulating and tumor infiltrating immunosuppressive myeloid cells and contribute to disease progression[14]. Additional studies of these cytokines/chemokines and other related immunomodulatory factors may provide greater insights into mechanisms of immunosuppressive myeloid cell accumulation during the tumor evolution/adaption, and reveal potential targets for preventing disease progression/drug-resistance. As clinical studies of anti-Her2 antibody and CDK4/6 inhibitor combination are still under way, the relevant data and samples of large patient cohorts are not yet available. Further investigation would be required to determine if immature myeloid cells infiltration is a general feature of disease progression and therapeutic resistance as demonstrated in this study.

In summary, this study supports the necessity and provides potential value to use single-cell profiling to trace, characterize, and resolve tumor evolution during the course of treatment, which could have a profound impact on future clinical decisions and rationally designed treatment strategies. Our preclinical

findings indicated that targeting immature myeloid cells subverts immunosuppressive TME and restores the vulnerability of highly aggressive breast cancer to checkpoint blockade immunotherapy. Along with on-going clinical trial and patient tissue biopsy, we envision that similar prospective in vivo resistance modeling and rational regimen design informed by tumor cells and TME alterations, could facilitate future translational precision medicine for cancer patients.

## Methods

**Animal model and syngeneic tumor transplantation.** FVB/N-MMTV-neu (202Mul) mice (Stock no: 002376) were purchased from Jackson Lab (Ben Harbor, ME). For tumor transplantation, treatment-resistant tumors were excised from MMTV-neu mice and immediately cut into small pieces of 3–5 mm in diameter. Donor tumors were transplanted into 4th mammary fat pad of MMTV-neu mice (12 to14-week old). Incisions were closed with wound clips which were removed after 7–10 days. Mice were monitored daily for tumor establishment. To establish/ensure the Ab + Pal resistant phenotype, the recipient mice were treated with Ab + Pal for 1–3 weeks after forming palpable tumors then indicated treatment was followed. Mouse experiments were performed in accordance with protocol approved by the University of Notre Dame IACUC committee.

**In vivo treatment.** Anti-HER2/neu antibody (clone 7.16.4, BE0277), mouse IgG2a Isotype control (Catalog# BE0085), anti-CTLA 4 antibody (clone 9H10, BE0131), anti-PD-1 antibody (clone RMP1-14, BE0146), anti-Ly6G/Ly6C (Gr1) antibody (clone RBC-8C5, BE0075), anti-CD3ε antibody (clone 145-2C11, BP0001), and polyclonal syrian hamster IgG (Catalog# BE0087) were purchased from BioXcell (West Lebanon, NH). Nulliparous female mice were enrolled for treatment when the spontaneous tumor reached a size of >500 mm³. Anti-HER2/neu antibody or the isotype IgG control was intraperitoneally administered at 10 mg/kg body weight in PBS twice weekly. Palbociclib isethionate salt (LC laboratories, P-7766) was prepared in 50 mM sodium lactate buffer and was given by oral gavage at a dose of 180 mg/kg every other day. Cabo (LC laboratories, C-8901) dissolved in 30% (v/v) propylene glycol, 5% (v/v) Tween 80, and 65% (v/v) of a 5% (w/v) dextrose solution in water, was orally administered at daily dose of 30 mg/kg. For ICB treatment, Gr1⁺ cells and T-cell depletion experiments, anti-PD-1, anti-CTLA 4, anti-Gr1, or anti-CD3ε antibodies (or their respective isotype IgG controls) were intraperitoneally administered at 200 μg per injection twice weekly, starting 1 day before anti-HER2/neu antibody and inhibitor treatment. The tumors were measured twice weekly using calipers. Tumor volume was calculated as length × width²/2. The volume of tumor when indicated treatment started was used as baseline for relative tumor volume calculation.

**Cell preparation.** Cells for single-cell RNA-seq were prepared by density centrifugation using Ficoll-Paque media (GE Healthcare, 17-5446-02) followed by magnetic-activated cell sorting (MACS)-based separation or enrichment. In brief, fresh mammary tumors were resected and minced with sterile scissors into ~1- to 2-mm³ pieces, then enzymatically digested in DMEM/F12 medium (10 ml/g tumor) containing 5% FBS, 2 mg/ml collagenase (Sigma), 0.02 mg/ml hyaluronidase (Sigma), and 0.01 mg/ml DNase I (Sigma) for 30 min at 37 °C with gentle agitation. Dissociated cells were centrifuged at 350 × g for 5 min with the brake on and discard supernatant. The pellet was resuspended with 3–5 ml of prewarmed TrypLE and incubated for 5 min. After adding 10 ml of DMEM/F12 medium supplemented with 2% FBS and passing through a 40-μm cell strainer (BD Biosciences), cells were centrifuged at 350 × g for 5 min and resuspended in MACS buffer [phosphate-buffered saline (PBS) with 0.5% bovine serum albumin (BSA) and 2 mM EDTA]. Cell suspension was carefully layered on top of 15 ml Ficoll-Paque media solution in a 50-ml Falcon tube and centrifuged at 1000 × g for 10 min at room temperature with the break off. The buffy layer at the interface was transferred and washed with cold MACS buffer. Following Ficoll separation, dead cells were eliminated by using the dead cell removal kit (Miltenyi Biotec, 130-090-101) per manufacturer's instruction. The live cell fraction was then incubated with CD45 magnetically labeled antibody (Miltenyi Biotec, 130-052-301) and passed through a LS magnetic column (Miltenyi Biotec, 130-042-401). The flow through fraction with enriched tumor cells (after depletion of CD45⁺ leukocytes) was collected. The cells retained in the column were then eluted as the isolated CD45⁺ TILs. For Gr1⁺ MDSC separation (using APR tumor transplantation model), the live cell fraction was subjected to a similar MACS-based isolation by application of the mouse MDSC isolation kit (Miltenyi Biotec, 130-094-538). Isolated cells were washed twice with cold MACS buffer and counted with a hemocytometer and diluted in cold PBS with 0.1% BSA and 2 mM EDTA at desired densities for Drop-seq.

**Drop-seq and sequencing analysis.** Single-cell transcriptomic profiles were generated using Drop-seq protocol[48]. Briefly, enriched tumor cell suspensions (pooled from three or four tumors) as prepared above were loaded on the microfluidic device (fabricated in-house, CAD file from McCarroll Lab website:

http://mccarrolllab.org/dropseq/) at ~100 cells/μl. CD45⁺ TILs and Gr1⁺ MDSCs were loaded at ~200 cells/μl (2 biological replicates for each treatment condition). Single-cell suspension and uniquely barcoded microbeads (Chemgenes, MACOSKO201110) suspended in lysis buffer were co-encapsulated in droplets by the microfluidic device. The droplets serve as compartmentalizing chambers for RNA capture. Once droplet generation was complete, collected droplets were disrupted and RNA-hybridized beads were harvested. Reverse transcription was performed using Maxima H Minus Reverse Transcriptase (Thermo Fisher Scientific, EP0752) with template switching oligo. cDNA was amplified and PCR products were then purified using AMpure Beads (Beckman Coulter). After quantification on a BioAnalayzer High Sensitivity Chip (Agilent), samples were fragmented and amplified for sequencing with the Nextera XT DNA sample prep kit (Illumina). The libraries were purified, quantified, and then sequenced on the Illumina HiSeq 2500 or NextSeq 500. Sequencing format was 25-cycle read 1, 8-cycle index 1, and 50-cycle read 2. Base calling was done by Illumina real time analysis (RTA) v1.18.64 and output of RTA was demultiplexed and converted to Fastq format with Illumina Bcl2fastq v1.8.4. Raw Drop-seq data was processed and aligned (STAR aligner) by following the standard Drop-seq pipeline. Briefly, reads were mapped to the mouse mm10 reference genome, then a digital gene expression data matrix was generated with counts of unique molecular identifiers (UMIs) for every detected gene (row) per cell barcode (column). We applied the knee plot method as recommended by the Drop-seq core computational protocol, which utilize the cumulative distribution of reads and identify an inflection point in the plot, to determine the number of cells (cell barcodes) represented in the expression matrix. Next, the Seurat R package[49] was used to perform data normalization, dimension reduction, clustering, and differential expression analysis. Cells from corresponding treatment groups were merged into a single matrix. For tumor cells (sequenced by Illumina HiSeq 2500), genes with detected expression in at least five cells were included and cells with either less than 600 genes and 1500 UMI or more than 4000 genes and 20,000 UMI were excluded. The percentage of reads aligned to mitochondrial genes per cells was calculated and cells with greater than 15% of transcripts derived from to mitochondrial genes[50] were filtered out. This resulted in 12,638 genes across 4817 cells. Potential contaminating stromal cells were further removed based on the expression of *Pdgfra* (marker for fibroblast), *Pecam*/CD31(marker for endothelial cells), CD45 and CD11b (markers for leukocytes). We finally obtained 4711 cells for further analysis. For TILs in Fig. 2b (sequenced by Illumina HiSeq 2500), genes with detected expression in at least two cells were included and cells with either less than 400 genes and 1200 UMI or more than 4000 genes and 30,000 UMI were excluded, and cells with greater than 10% of transcripts derived from to mitochondrial genes were removed. For Fig. 2g (sequenced by NextSeq 500), genes with detected expression in at least two cells were included, cells with either less than 500 genes and 1500 UMI or more than 5000 genes and 50,000 UMI were excluded, and cells with mitochondrial genes greater than 10% were also removed. For Fig. 4b (sequenced by NextSeq 500), genes with detected expression in at least ten cells were included, cells with either less than 400 genes or more than 5000 genes were excluded, and cells with mitochondrial genes greater than 10% were also removed. The filtered matrix was scaled to 10,000 molecules and log-normalized per cell to correct for the difference in sequencing depth between single cells.

**Gene set enrichment analysis.** ssSEA[51] was run using GSVA v1.28.0 in R using single-cell expression matrix with UMI values. We applied hallmark gene sets and canonical pathways from KEGG, REACTOME, and BIOCARTA gene sets of the C2 collection of Molecular Signatures Database (MSigDB) (converted to mouse gene symbols) to each single cell to obtain enrichment score for each signature. Our custom and experimentally generated MDSCs signature was based on marker genes (top 300 differential expressed genes) of cell clusters by Seurat package. Drug target genes of FDA-approved small molecular PKIs were adapted from The Blue Ridge Institute for Medical Research (http://www.brimr.org/PKI/PKIs.htm).

**Mass cytometry by time of flight (CyTOF).** Fresh or cryopreserved mammary tumors were enzymatically digested followed by density centrifugation and dead cell removal as aforementioned. Cells for CyTOF were washed and resuspended in Maxpar PBS (Fluidigm, 201058). Cells suspensions were incubated with Cell-ID Cisplatin (Fluidigm, 201064) for 5 min and then washed in Maxpar Cell Staining Buffer (Fluidigm, 201068). FC receptors were blocked by incubation with TruStain fcX in 100 μl MaxPar Cell Staining Buffer for 15 min at room temperature. Cells were incubated with a cocktail of CyTOF antibodies (Supplementary materials) for 30 min at room temperature and then washed in MaxPar Cell Staining Buffer. Optimal concentrations were determined for each antibody by titration. Cells were incubated with Cell-ID Cisplatin (Fluidigm, 201064) at 2.5 μM for 2.5 min for viability staining. Cells were resuspended and fixed in 1.6% PFA prepared in MaxPar PBS for 20 min and then Intercalator (Fluidigm, 201192B) dissolved in MaxPar Fix and Perm Buffer (Fluidigm, 201067) for 1 h or overnight at 4°C. Following nuclear labeling, cells were washed once in MaxPar Cell Staining Buffer and twice in MaxPar Water (Fluidigm, 201069). Samples were brought to 500,000 particulartes/ml in MilliQ water containing 0.1× EQ beads (Fluidigm, 201078) and run in 450 μl injections on a CyTOF2 instrument. CyTOF data were analyzed and visualized using Cytobank Premium (Cytobank, Inc).

**Statistical analysis**. Statistical tests were performed in GraphPad Prism version 7.0 or in R. Data were analyzed with two-tailed unpaired Student's $t$ tests when comparing means of two groups and one-way ANOVA when comparing more than two groups. Chi-square test was used to compare the proportion of cells. Survival curves were compared with the log-rank (Mantel–Cox) test. $P$ values < 0.05 were considered significant.

## Data availability

The single-cell RNA sequencing data have been deposited in the GEO data repository under accession number GSE122336. All the other data supporting the findings of this study are available within the article and its supplementary information files and from the corresponding author upon reasonable request. A reporting summary for this article is available as a Supplementary Information file.

## Code availability

Computational analyses were performed in R (version 3.5.0). The code used for processing raw data of Drop-seq is available at http://mccarrolllab.org/dropseq/. Seurat package (V2.3.2) is available at https://satijalab.org/seurat/ and GSVA (V1.28.0) package is available at https://www.bioconductor.org/packages/release/bioc/html/GSVA.html. The scripts used for the described analysis are available from the corresponding authors upon reasonable request.

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

## Acknowledgements

We would like to thank Zhang Lab members for scientific insights and support. For insightful technical assistance, we thank Jacqueline Lopez, M.S., Samuel W. Brady, Ph.D., Andrea Gunawan, M.S., and Charles R. Tessier, Ph.D. We are additionally grateful for the use of the following core facilities: Notre Dame Genomics and Bioinformatics Core Facility, Notre Dame Freimann Life Sciences Center, Harper Cancer Research Institute Biorepository, Indiana University Simon Cancer Center Core Facility, Michigan State University Genomics Core Facility, and Indiana University School of Medicine-South

Bend Imaging and Flow Cytometry Core Facility. This work was partially funded by NIH R01 CA194697-01 (S.Z.), NIH R01 CA222405-01A1(S.Z.), Notre Dame CRND Catalyst Award (S.Z. and I.H.G.), NIH CTSI core facility pilot grants (S.Z.), U54 pilot grant U54CA209978 (S.Z.), Notre Dame ADT grant (S.Z.), NIH CTSI Postdoc Challenge Award (Q.W.). We would additionally like to acknowledge and thank the Dee Family endowment (S.Z.).

## Author contributions

Q.W. and S.Z. conceived the original hypothesis and designed experiments. Q.W., I.H.G., S.M.G., L.S. and J.A.H. performed experiments. Q.W. and S.Z. analyzed data. X.L. provided direction and guidance to this study. Q.W. and S.Z. wrote and revised the manuscript. S.Z. supervised the study.

## Additional information

**Competing interests:** The authors declare no competing interests.

**Peer Review Information:** *Nature Communications* thanks the anonymous reviewers for their contribution to the peer review of this work. Peer reviewer reports are available.

