## [Peer Review File · Nature Communications]

Reviewers' comments:

Reviewer #1 (Remarks to the Author): Expert in breast cancer and immunology

This report uses single cell profiling by RNA-seq and CyTOF to prospectively model the evolution of the TME of HER-2+ breast cancer in MMTV-neu mice treated with CDK4/6 inhibitors combined with trastuzumab-like monoclonal antibodies. The authors identify an immunosuppressive myeloid cell population in the treatment-resistant TME, and then demonstrate that the combination of the TKI cabozatinib combined with immune checkpoint blockade enhances antitumor immunity and overcomes resistance. The work is very interesting and demonstrates the highly effective use of mouse modeling to understand mechanisms of treatment response and resistance, which then guides the development of the best therapies for the evolving biology. Strengths of the work include the use of state-of-the art technology to thoroughly evaluate the evolution of the TME under therapeutic pressure of drugs under active clinical investigation in the clinic, and the examination of the impact of treatment sequencing on the antitumor activity (a variable in combinations therapies that is understudied). The work is carefully and systematically laid out and the results are convincing. A few questions:

1. In the middle of the first page of results the authors state that treatment sensitive disease is residual disease 10-14 days post 7.16.4 and palbo therapy. Is this really treatment sensitive disease, or has the selection of treatment resistant disease already begun? What would you get if you conduct an analysis within a very short time after beginning the therapy, such as 24-72 hours?
2. In the sequential treatments toward the end of the paper, the authors show that the therapy can be given with a significant antitumor effect, and then stopped, at which time the tumors grow out, and then re-started with the re-induction of response. This can be observed at least three times. This may be a little curious if T cells, the target of immune checkpoint blockade, are involved as there may be some element of immune memory. The authors have not presented T cell depletion studies to demonstrate a dependence on T cells with these therapies, only depletion of MDSC with Gr1. This should be discussed.
3. The addition of cabozatinib to 7.16.4 restores sensitivity to treatment resistant tumors. Monoclonal antibodies may work by many mechanisms, including ADCC and the induction of adaptive immunity. Is the mechanism underlying this limited to modulation of MDSC?

Reviewer #2 (Remarks to the Author): Expert in CyTOF

This high-quality study entitled, “Single-cell profiling guided combinatorial immunotherapy for fast-evolving CDK4/6 inhibitor resistant HER2-positive breast cancer”, utilizes single-cell approaches (Drop-Seq/CyTOF) to decipher microenvironmental influences on resistance to anti-Her2/Neu antibody + CDK4/6 inhibitor treatment. By identifying the cell type (IMCs-MDSCs) that directs resistance, the authors leverage signaling mechanisms within these cells to overcome resistance (a TKI inhibitor + immunotherapy). Overall, this is a translationally relevant study utilizing an interesting pre-clinical model and data to decipher non-cell autonomous mechanisms of resistance.

1. Problem with terminology. The authors utilize the term “resistance” in many contexts. However, some of the uses are wrong, for example, “treatment resistant tumor cells”. In this case, it seems that it is the microenvironment conditioning the resistance, and not the tumor cells being intrinsically resistant.
2. The major issue of this paper arises from the use of single-cell data coming from a single tumor (or the number of tumors queried by single-cell analysis is unclear). It is hard press for this reviewer to accept that multiple independent tumors from different mice arrive at the exact same resistance mechanism to the treatment. While it is possible that this mechanism is universal, it is not explicitly shown in the paper. Using orthogonal marker-based approaches to demonstrate this mechanism occurring multiple tumors is required.
3. Some of the statements regarding scRNA-seq are over-interpreted. For example, the “mixed single cells” defined as “seeds” can just happen to be stochastic noise. These cells were never proven as “seeds” in the paper. These statements should be moderated.
4. Overreliance on single-cell RNA-seq data. Given the lack of replicates, the majority of conclusions drawn about gene expression comes strictly from sequencing data. Validation using other approaches such as flow sorting and then RT-PCR is required.
5. Lack of replicates and statistics throughout the paper. Examples include Figure 3B (no error bars), plots of tumor volume increases (no stats), etc.
6. Tumor volume – would be nice to see an image of tumors that are progressing and those that regressing in the supplement- and visually how these measurements are performed. It is unclear what the effect sizes are.
7. TILs in figure 2 were visualized using CD3. A CD8 stain will add to the anti-tumor nature of these cells.
8. Unclear why during combination therapy (with cabo or icb) that Pal is dropped. Please explain.

Reviewer #3 (Remarks to the Author): Expert in single cell sequencing

In the manuscript entitled “Single-cell profiling guided combinatorial immunotherapy for fast-evolving CDK4/6 inhibitor resistant HER2-positive breast cancer” the authors characterize a syngeneic mouse breast cancer model that is resistant to anti-Her2/Neu antibody and Palbociclib (Ab+Pal) combination treatment, and develop novel therapeutic strategies to overcome resistance. Their model was derived by long-term treatment with the two drugs (Ab+Pal), although an initial response to the drug combination was observed. These observations of an initial response and eventual resistance are in line with clinical observations in patients treated with targeted therapies. To characterize the changes induced upon development of resistance, the authors used state-of-the-art high-throughput single-cell analyses of tumor cells and leukocytes. Interestingly, the authors identified a distinct immunosuppressive immature myeloid cell (ICM) population in the treatment resistant tumors using these single-cell analyses. Depletion of Gr1+ MDSCs, which formed the majority of the ICM population, inhibited growth of resistant tumors. Importantly, the authors also identified a combination treatment strategy involving ICM-targeting tyrosine kinase inhibitor Cabozantinib and immune checkpoint blockade that could overcome resistance to the initial Ab+Pal treatment by subverting the immunosuppressive tumor microenvironment in the resistant tumors. Their results provide a compelling argument for using combinatorial immunotherapy for HER2+ breast cancer, particularly after resistance to standard therapy has emerged.

Major points:

1. MACS beads will partially deplete/enrich for CD45+ cells from the dissociated tumor samples; however, this is neither complete nor does it eliminate other stromal cells that are CD45negative. First, it would be important to determine how well the enrichment of tumor cells and CD45+ cells worked. It is not clear whether the authors excluded contaminating cells from their analyses. Second, especially in the tumor scRNAseq experiment, it would be useful to determine how many cells are stromal cells and how this composition changes depending on treatment. These analyses could be done from the scRNAseq data and confirmed by another method such as immunostaining or western blot.
2. While the shift in the immune microenvironment between control, sensitive, and resistant tumors has been investigated using several methods (scRNAseq, IHC, CyTOF), the extent of this shift seems to vary between the different methods. For instance, their scRNAseq data suggest T&NK cells make up 50% of CD45+ cells in APS and only about 10% of Control and APR tumors. However, using CyTOF these cell populations make up less than 20% of the CD45+ cells in APS and about 10% in the other two tumors. While different methods may give different results due to their inherent differences such as the number and types of markers used for CyTOF, the authors should comment on these discrepancies and address them if possible. The authors should also include their gating strategy for the CyTOF experiment.
 - a. The authors should confirm their results with more extensive IHC (or IF) staining of tumors using markers for different immune cells. They have only performed this for CD3+ T cells, but not on

macrophages, immature myeloid cells and the other cell types they have identified. This would also allow the authors to examine the localization of the different immune cells within the tumor/at the tumor stroma interface and determine whether they vary by treatment.

b. This shift is also based on relative numbers to CD45+ cells. However, do any of the treatment change the number of infiltrating CD45+ cells? IHC analyses can give more information on the total numbers of infiltrating immune cell populations.

c. Apart from the types of immune cell populations, it would be useful if the authors could examine the activation status and/or phenotype of some of the major populations. For instance, Granzyme B expression in CD8+ T cells or M1/M2 phenotypes of macrophages, and how they differ between treatment groups.

3. The set-up of experiment shown in Fig 3G is not entirely clear. Were CD45+ cells first enriched from the tumor and then Gr1+ cells enriched from the CD45+ fraction? Otherwise, we would expect Gr1+ cells to be a very big part of the CD45+ population, especially because APR tumors were analyzed for this experiment (around 80% based on the authors previous scRNAseq in Fig 2C). Similarly, group I cells would not show as high expression of macrophage and T/NK cell genes in the total CD45+ population.

4. Confirm activation of T cell anti-tumor activity observed in scRNAseq in Figure 4G with other methods, such as IHC of granzyme B.

Minor points:

1. The authors used Ab+Pal treatment to obtain resistant tumors. In the text and several figures (e.g. Fig 1B, Fig 1H, Fig 2K), the authors refer to the continued treatment as “days post treatment”. This should probably be “days of treatment” or similar, as treatment was continued and not stopped.

2. It would be more better if the authors included actual p-values instead of asterisks in all their figures.

3. Include p-values for Figures 2K, 3D, 5F, S7A. Include p-values for CyTOF data (Figures 2E, 4A).

4. Similarly, it would be useful if the authors included Hazard ratios of their Kaplan-Meier curves in Figures S1B, 3F, 5B.

5. Figure legends should include number of animals/group used (Figures S1B, 3F, 5B)

6. Spelling in Figures 1B & S1C: Volume instead of Volumn

7. Gene names should be italicized, such as in Figures S2D&H, S3B-E, S5C

8. Check colors in Figure S3C, figure and legend do not correspond

9. It would be helpful for the reader if the authors described in a bit more detail the markers used particularly in the CyTOF experiment in Fig 2E. They do so partially later in the text, but it's

recommended that any marker mentioned for the first time is explained briefly as well as the rationale for using this marker. Graphs with markers should have an explanation what the marker is for, so readers not familiar with immunology can quickly understand the context.

10. It would be helpful for the reader if t-SNE plots in the main figures would include the group numbers, so the reader does not have to look those up in the supplemental figures.

Reply to Reviewers' Comments

Re: Manuscript # NCOMMS-19-00732

We would like to thank the editor and the reviewers for valuable and constructive comments. We very much appreciate that the reviewers' supportive comments including “The work is carefully and systematically laid out and the results are convincing.” (Reviewer #1); “This high-quality study” and that “a translationally relevant study utilizing an interesting pre-clinical model” (Reviewer #2); “Their results provide a compelling argument for using combinatorial immunotherapy for HER2+ breast cancer, particularly after resistance to standard therapy has emerged.” (Reviewer #3). In this revised manuscript, we have carefully addressed each of the reviewers' comments in full by adding a significant amount of new data (~10 new panels), additional descriptions of Methods (and updated Supplementary Methods) and new section of Discussion. Here, we have indicated the changes made to the manuscript as blue text to take into account the comments from the three reviewers, which are reproduced below in *italics* (font size 10) for your easy reference. Our point-by-point response to comments from the reviewers are as follows:

Reviewer #1 (Remarks to the Author): Expert in breast cancer and immunology

This report uses single cell profiling by RNA-seq and CyTOF to prospectively model the evolution of the TME of HER-2+ breast cancer in MMTV-neu mice treated with CDK4/6 inhibitors combined with trastuzumab-like monoclonal antibodies. The authors identify an immunosuppressive myeloid cell population in the treatment-resistant TME, and then demonstrate that the combination of the TKI cabozatanib combined with immune checkpoint blockade enhances antitumor immunity and overcomes resistance. The work is very interesting and demonstrates the highly effective use of mouse modeling to understand mechanisms of treatment response and resistance, which then guides the development of the best therapies for the evolving biology. Strengths of the work include the use of state-of-the art technology to thoroughly evaluate the evolution of the TME under therapeutic pressure of drugs under active clinical investigation in the clinic, and the examination of the impact of treatment sequencing on the antitumor activity (a variable in combinations therapies that is understudied). The work is carefully and systematically laid out and the results are convincing. A few questions:

Reply: We thank reviewer’s positive comments about our work.

1. In the middle of the first page of results the authors state that treatment sensitive disease is residual disease 10-14 days post 7.16.4 and palbo therapy. Is this really treatment sensitive disease, or has the selection of treatment resistant disease already begun? What would you get if you conduct an analysis within a very short time after beginning the therapy, such as 24-72 hours?

Reply: We thank the reviewer for this point. Indeed, tumor shrinkage was observed after 48-72 hours of 7.16.4 Ab+Pal treatment. There was a clear drug selection pressure imposed on the tumors shortly after the treatment began. While being responsive to Ab+Pal treatment at the beginning, tumor evolves. It is difficult to truly tease out sensitive vs residual tumor cells at the molecular level from the evolving tumors, even we conduct the analysis after a short-term (24-72 hours) treatment. We agree with the reviewer’s concern that the terminology of “sensitive” tumor might not precisely reflect the nature of the tumor. After consulting a medical oncologist at our institution, we believe the term “**Ab+Pal resPonsive (APP)**” nomenclature is a more precise term to describe the tumor collected after 10-14 days of Ab+Pal

treatment. Thus, in this revised manuscript, we have replaced “APS” with “APP” throughout the text and figures.

2. In the sequential treatments toward the end of the paper, the authors show that the therapy can be given with a significant antitumor effect, and then stopped, at which time the tumors grow out, and then re-started with the re-induction of response. This can be observed at least three times. This may be a little curious if T cells, the target of immune checkpoint blockade, are involved as there may be some element of immune memory. The authors have not presented T cell depletion studies to demonstrate a dependence on T cells with these therapies, only depletion of MDSC with Gr1. This should be discussed.

Reply: Thank you for this insightful comment. The Ab+Cabo treatment indeed increased T-cell infiltration (**Fig.R1A**, also shown in original Fig.S9C). Inspired by the reviewer’s comment, we next performed T-cell depletion studies during Ab+Cabo treatment against Ab+Pal resistant (APR) tumors. We found that T-cell depletion resulted in a significant reduction of the therapeutic efficacy of Ab+Cabo treatment (**Fig. R1B**). This result indicates that the optimal therapeutic activity of Ab+Cabo against APR tumors is dependent on T cells. We have re-organized Fig.3 and updated the Results part accordingly in the revised manuscript to incorporate these findings (Please see Page 10, 2nd paragraph, in blue text).

Fig.R1 (A) Ab+Cabo treatment increased T-cell infiltration; (B) T-cell depletion resulted in reduction of tumor suppression by Ab+Cabo against Ab+Pal resistant tumors.

Accordingly, we have added “This long-lasting disease control by such treatment might also engage or enhance immune memory, considering the optimal therapeutic activity of Ab+Cabo with a dependence on T cells.” in the Discussion section of the revised manuscript (Please see Page 15, 2nd paragraph, in blue text).

3. The addition of cabozantinib to 7.16.4 restores sensitivity to treatment resistant tumors. Monoclonal antibodies may work by many mechanisms, including ADCC and the induction of adaptive immunity. Is the mechanism underlying this limited to modulation of MDSC?

Reply: We agree with the reviewer and it has been well recognized that the activity of antibody-based targeted-therapies depends on immune-mediated mechanisms. In our manuscript, we didn’t intend to exclude other potential mechanisms as we have previously stated in the discussion "In the present study, although cabozantinib-containing regimen inhibit immature myeloid cells in the APR tumor, we could not exclude the possible direct effect of cabozantinib on tumor cells and other stroma compartments".

We want to emphasize that our study did demonstrate that the infiltration of MDSC/IMCs is a significant mechanism directly leading to resistance and modulation of IMC/MDSCs by cabozantinib sensitizes the resistance tumors to ICB treatment.

Reviewer #2 (Remarks to the Author): Expert in CyTOF

This high-quality study entitled, “Single-cell profiling guided combinatorial immunotherapy for fast-evolving CDK4/6 inhibitor resistant HER2-positive breast cancer”, utilizes single-cell approaches (Drop-Seq/CyTOF) to decipher microenvironmental influences on resistance to anti-Her2/Neu antibody + CDK4/6 inhibitor treatment. By identifying the cell type (IMCs-MDSCs) that directs resistance, the authors leverage signaling mechanisms within these cells to overcome resistance (a TKI inhibitor + immunotherapy). Overall, this is a translationally relevant study utilizing an interesting pre-clinical model and data to decipher non-cell autonomous mechanisms of resistance.

1. Problem with terminology. The authors utilize the term “resistance” in many contexts. However, some of the uses are wrong, for example, “treatment resistant tumor cells”. In this case, it seems that it is the microenvironment conditioning the resistance, and not the tumor cells being intrinsically resistant.

Reply: We thank the reviewer for the excellent point. We have checked the uses of “resistance” and changed accordingly in our revised manuscript.

2. The major issue of this paper arises from the use of single-cell data coming from a single tumor (or the number of tumors queried by single-cell analysis is unclear). It is hard press for this reviewer to accept that multiple independent tumors from different mice arrive at the exact same resistance mechanism to the treatment. While it is possible that this mechanism is universal, it is not explicitly shown in the paper. Using orthogonal marker-based approaches to demonstrate this mechanism occurring multiple tumors is required.

Reply: Thank you for the comment and important suggestion. We apologize for the lack of explicitly in the manuscript regarding the number of tumors sequenced. We provided the information regarding this point in the GEO data repository (accession number GSE122336). Tumor cell (CD45- cells) sequencing were conducted by pooling cells from three or four tumors. “For tumor-infiltrating leukocyte (TIL) and MDSC samples: Rep1 and Rep2 represented two biological repeats. The last three number in name of raw files (001 and 002) for some TIL samples as indicated, represented technical repeat.”

We have now included this information in the “Drop-seq and sequencing analysis” paragraph of “Material and Methods” part in our revised manuscript (blue text), indicating that enriched tumor cell suspensions were pooled from three or four tumors, and two biological replicates for each treatment condition were sequenced for CD45+ TILs and Gr1+ cells.

Per the reviewer’s suggestion, in this revised manuscript, we performed IF staining for MDSCs using CD11b and Gr1 antibodies in frozen tumor sections. As shown in **Fig.R2 (Supplementary Fig.5F** in the revised manuscript), we observed significantly more MDSCs (CD11b and Gr1 double positive cells) infiltration in the APR tumors compared to control and APS tumors (**APP** in the revised manuscript. Please refer to our answer to Reviewer 1’s comment 1), indicating it is universal that immature myeloid cells are enriched in APR tumors, which contributes to the mechanisms of Ab+Pal resistance.

Fig.R2 IF staining of tumor infiltrated MDSC/IMCs.

3. Some of the statements regarding scRNA-seq are over-interpreted. For example, the “mixed single cells” defined as “seeds” can just happen to be stochastic noise. These cells were never proven as “seeds” in the paper. These statements should be moderated.

Reply: We thank the reviewer for this helpful comment. We have removed these statements accordingly (Page 5).

4. Overreliance on single-cell RNA-seq data. Given the lack of replicates, the majority of conclusions drawn about gene expression comes strictly from sequencing data. Validation using other approaches such as flow sorting and then RT-PCR is required.

Reply: Thank you for this comment. Per reviewer’s suggestion, we conducted additional validations of our findings.

- Validation the expression of Lcn2 and Mgst1 using flow sorting and then RT-PCR

In the original manuscript, we found that Lcn2 and Mgst1 are two marker genes of MDSCs/IMCs by scRNA-seq analysis. As suggested, T cells (CD3+), macrophages (F4/80+) and MDSCs/IMCs (Gr1+) were sorted out from APR tumors then quantitative PCR was performed. Indeed, as shown in **Fig. R3A (Supplementary Fig. 6D)** in the revised manuscript), compared to T cells and macrophages, significantly higher expression levels of both Lcn2 and Mgst1 were observed in Gr1+ MDSCs/IMCs.

- Validation the expression of Kit and Met by RT-PCR

As shown in **Fig. R3B (Supplementary Fig. 7G)** in the revised manuscript), we found that CD45+TILs from APR tumors had higher levels of Kit and Met compared to those of from either Ctrl tumors or APS tumors. In addition, as shown in **Fig. R3C (Supplementary Fig. 7H)** in the revised manuscript) among the 3 cell types assorted, Gr1+ population showed the highest expression of Kit and Met.

Fig.R4 (A) Validation the expression of Lcn2 and Mgst1 and (B, C) validation the expression of Kit and Met by using flow sorting and then RT-PCR.

Collectively, our major findings derived from single-cell RNA-seq have been validated by flow sorting and then RT-PCR. We have updated the Results part accordingly (Please see Page 8 last 4 lines and Page 8 1st paragraph, blue text) to incorporate these data.

5. Lack of replicates and statistics throughout the paper. Examples include Figure 3B (no error bars), plots of tumor volume increases (no stats), etc.

Reply: Thank you for the comment. To clarify, regarding tumor volume plots (Figure 1H, Figure 2K, Figure 3D&E, Figure 5D&F), each line represents one tumor. In this revised manuscript, the number of animals used in each group and the statistics in these plots have been included. Please see the updated Figure 1H, Figure 2K, Figure 3G-I, Figure 5D&F.

Figure 3B (right) showing the percentage of Kit or/and Met-expressing cells among sequenced cells as shown in t-SNE plot (Figure 3B left). Following the reviewer's suggestion, we have included the p values (by pairwise comparisons of three-sample proportion test) in the revised figure.

6. Tumor volume – would be nice to see an image of tumors that are progressing and those that regressing in the supplement- and visually how these measurements are performed. It is unclear what the effect sizes are.

Reply: We thank the reviewer for the suggestion. We have added images of tumors (**Fig.R5** as shown below and in new **Supplementary Fig. 1D**) and provided the relative tumor volumes at the indicated time (days of treatment). Relative tumor volume = tumor volume at the indicated time (days of treatment) / volume at the start point of treatment.

Fig.R5 Representative images of the tumor. RV: relative tumor volume. Scale bar, 1 cm.

In our study, the indicated treatment was started when the spontaneous mammary tumor volume reached $> 500\text{mm}^3$. As shown in **Figure 5D**, the therapies were effective for even very late stage tumors (2 mice with tumor volumes $> 2000\text{mm}^3$ and 3 mice with tumor volumes $1000\sim 2000\text{mm}^3$).

7. TILs in figure 2 were visualized using CD3. A CD8 stain will add to the anti-tumor nature of these cells.

Reply: Thank you for this helpful suggestion. As shown in **Fig. R6**, we have added the result of CD8 staining in the updated **Figure 2E**.

Fig.R6 IF staining for CD3 and CD8 in Ctrl, APP and APR tumors.

8. Unclear why during combination therapy (with cabo or icb) that Pal is dropped. Please explain.

Reply: Thanks for this important question. In our studies, despite a promising initial response, acquired resistance emerged rapidly to 7.16.4 Ab+ CDK4/6 inhibitor Palbociclib treatment (Fig.1 A&B). Our results showed that increased antigen presentation and stimulate IFN signaling (Fig.1F&G), suggesting elevated tumor immunogenicity in Ab+Pal treated resistant tumors. On the other hand, however, long-term 7.16.4 Ab+Pal treatment led to a significant increase of immunosuppressive immature myeloid cells in the TME, which in turn diminished the efficacy of ICB (Fig. 1H). Guided by single-cell transcriptome analysis, ICM-targeting tyrosine kinase inhibitor cabozantinib-containing combinatorial immunotherapy could effectively inhibit Ab+Pal resistant tumors (Fig.3), by reducing immunosuppressive immature myeloid cells and enhancing anti-tumor immunity (Fig.4).

These results prompted us to hypothesize that a short period of Ab+Pal treatment will be sufficient to primes the anti-tumor immunity before the emergence of resistance. Considering a combination of four different treatments (Ab+Pal+Cabo+ICB) simultaneously is clinically challenging to execute and might lead to undesired toxicity, we designed sequential administration of Ab+Cabo or Ab+Cabo+ICB combination after a short period of Ab+Pal. Indeed, we demonstrated that such sequential combinatorial immunotherapy is sufficient to control the tumor progression and enabled a sustained response of rapidly evolving HER2/neu-positive breast cancer.

Reviewer #3 (Remarks to the Author): Expert in single cell sequencing

In the manuscript entitled “Single-cell profiling guided combinatorial immunotherapy for fast-evolving CDK4/6 inhibitor resistant HER2-positive breast cancer” the authors characterize a syngeneic mouse breast cancer model that is resistant to anti-Her2/Neu antibody and Palbociclib (Ab+Pal) combination treatment, and develop novel therapeutic strategies to overcome resistance. Their model was derived by long-term treatment with the two drugs (Ab+Pal), although an initial response to the drug combination was observed. These observations of an initial response and eventual resistance are in line with clinical observations in patients treated with targeted therapies. To characterize the changes induced upon development of resistance, the authors used state-of-the-art high-throughput single-cell analyses of tumor cells and leukocytes. Interestingly, the authors identified a distinct immunosuppressive immature myeloid cell (ICM) population in the treatment resistant tumors using these single-cell analyses. Depletion of Gr1+ MDSCs, which formed the majority of the ICM population, inhibited growth of resistant tumors. Importantly, the authors also identified a combination treatment strategy involving ICM-targeting tyrosine kinase inhibitor Cabozantinib and immune checkpoint blockade that could overcome resistance to the initial Ab+Pal treatment by subverting the immunosuppressive tumor microenvironment in the resistant tumors. Their results provide a compelling argument for using combinatorial immunotherapy for HER2+ breast cancer, particularly after resistance to standard therapy has emerged.

Major points:

1. MACS beads will partially deplete/enrich for CD45+ cells from the dissociated tumor samples; however, this is neither complete nor does it eliminate other stromal cells that are CD45negative. First, it would be important to determine how well the enrichment of tumor cells and CD45+ cells worked.

Reply: Thank you for the comments and suggestion. Magnetic-activated cell sorting (MACS) is a well-established and widely used method for cell separation. To determine how well the MACS method actually worked for our application, we checked the purity of harvested cells after MACS separation by flow cytometry. As shown in **Fig.R7A**, in the case of CD45+ depletion, the frequency of EpCAM+ cells increased to > 98% after depletion of CD45+ cells by MACS. In the case of enrichment of CD45+ cells by MACS, > 90% of cells were CD45+ as determined by flow cytometry (**Fig.R7B**). These results demonstrated that the depletion and enrichment of CD45+ cells by MACS achieved more than 90% efficiency.

Fig.R7 Determine the purity of MACS method enriched tumor cells (A) and TILs (B) by flow cytometry.

It is not clear whether the authors excluded contaminating cells from their analyses. Second, especially in the tumor scRNAseq experiment, it would be useful to determine how many cells are stromal cells and how this composition changes depending on treatment. These analyses could be done from the scRNAseq data and confirmed by another method such as immunostaining or western blot.

Reply: We would like to thank the reviewer for this helpful suggestion. In our original manuscript, we analyzed the single cell transcriptomes of enriched tumor cells by clustering (after QC). The unbiased clustering did not show obvious stromal cell subsets. Following the reviewer's comment, we reanalyzed our scRNAs-seq data to exclude the potential contaminating cells. First, we evaluated the expression of stromal cells related marker genes including *Pdgfra* (a marker for fibroblast), *Pecam* (CD31) (a marker for endothelial cells), *Ptpnc* (CD45) and *Itgam* (CD11b) (markers for leukocytes). As shown in **Table R1**, based on fibroblasts, endothelial cells and leukocytes markers expression, we identified total 106 contaminated cells among total 4817 single cells recovered. In this revised manuscript, those 106 cells have been removed from our new analyses.

Table R1 Characterization of sequenced enriched tumor cells.

Treatment	Cells recovered	Tumor cells	Endothelial cells (Pecam/CD31+)	Fibroblasts (Pdgfra+)	Leukocytes (CD45+ and/or CD11b+)
Ctrl	1978	1914 (96.76%)	9 (0.455%)	9 (0.455%)	46 (2.326%)
Ab	803	790 (98.38%)	NA	7 (0.872%)	6 (0.747%)
Pal	1119	1106 (98.84%)	10 (0.894%)	3 (0.268%)	NA
APS	160	153 (95.625%)	NA	4 (2.5%)	3 (1.875%)
APR	757	748 (98.81%)	NA	NA	9 (1.19%)
Total	4817	4711 (97.78%)	19 (0.394%)	23 (0.477%)	64 (1.33%)

Next, we reanalyzed tumor scRNA-seq data without those 106 contaminated cells. The new tSNE plots generated by the new analyses (updated **Fig. 1D**) showed very subtle difference compared to the original version. And the updated **Fig. 1E-G** showed no visible change compared to the original ones. We have now updated all related figure panels (including **Fig.1D-G** and **supplementary Fig.2**) and Methods section accordingly in our revised manuscript. Finally, since the non-epithelial cells only account for about 2.22% of total cells sequenced/recovered in the tumor scRNA-seq experiment, we do not feel further validation using IHC or western blot will provide a significant information to our data interpretation.

2. While the shift in the immune microenvironment between control, sensitive, and resistant tumors has been investigated using several methods (scRNAseq, IHC, CyTOF), the extent of this shift seems to vary between the different methods. For instance, their scRNAseq data suggest T&NK cells make up 50% of CD45+ cells in APS and only about 10% of Control and APR tumors. However, using CyTOF these cell populations make up less than 20% of the CD45+ cells in APS and about 10% in the other two tumors. While different methods may give different results due to their inherent differences such as the number and types of markers used for CyTOF, the authors should comment on these discrepancies and address them if possible.

Reply: We appreciate the reviewer for this comment. As the reviewer pointed out, “*different methods may give different results due to their inherent differences*”, instead comparing the absolute quantitative percentage numbers between different methods, in our study, defining the relative abundance (e.g. between responsive and resistant phenotypes) in the same experiment using the same method is more meaningful. Despite certain discrepancies, the observations (by scRNAseq, IHC/IF, CyTOF) collectively revealed that APS (now annotated as “APP” in response to the reviewer 1’s suggestion) tumors contain more infiltrated T and NK cells. In contrast, APR TME were dominated by IMCs.

The authors should also include their gating strategy for the CyTOF experiment.

Reply: As the reviewer suggested, we have now included the gating strategy for the CyTOF as shown in new **Supplementary Fig.5** and **Supplementary Fig.9**.

a. The authors should confirm their results with more extensive IHC (or IF) staining of tumors using markers for different immune cells. They have only performed this for CD3+ T cells, but not on macrophages, immature myeloid cells and the other cell types they have identified. This would also allow the authors to examine the localization of the different immune cells within the tumor/at the tumor stroma interface and determine whether they vary by treatment.

Reply: We thank the reviewer for this suggestion. In this revised manuscript, we have validated the result of MDSC/IMCs infiltration by IF staining using CD11b and Gr1 antibodies in frozen tumor sections (**Fig.R2**, as described in response to reviewer 2, and **Supplementary Fig. 5F** in the revised manuscript). We did not observe obvious enrichment of immune cells at the tumor-stroma interface. When focusing our analysis on the immune cells infiltrated within the tumor mass, we observed that immature myeloid cells (CD11b+Gr1+) are significantly enriched in APR tumors compared to control or APS (APP in revised Figure) tumors (**Fig. R2**).

Fig.R2 IF staining of tumor infiltrated MDSC/IMCs.

b. This shift is also based on relative numbers to CD45+ cells. However, do any of the treatment change the number of infiltrating CD45+ cells? IHC analyses can give more information on the total numbers of infiltrating immune cell populations.

Reply: We agree with the reviewer. We performed CD45 IHC staining and found that more CD45+ cells in both APS (APP in the revised manuscript) and APR tumors compared to Ctrl (**Fig. R8**, new **Supplementary Fig. 3** in the revised manuscript). This result has been incorporated in the result section accordingly (Please see page 6, blue text).

Fig.R8 IHC staining of tumor infiltrated CD45+ cells.

c. Apart from the types of immune cell populations, it would be useful if the authors could examine the activation status and/or phenotype of some of the major populations. For instance, Granzyme B expression in CD8+ T cells or M1/M2 phenotypes of macrophages, and how they differ between treatment groups.

Reply: Thank you for this helpful suggestion. In conjunction with suggestions by Reviewer #2, as shown in **Fig.R6**, we have added the result of CD8 staining in the updated **Figure 2E** (and **Supplementary Fig. 10C** as well for Ab+Cabo treatment).

Fig.R6 IF staining for CD3 and CD8 in Ctrl, APS (APP in the revised manuscript) and APR tumors.

Regarding M1/M2 phenotypes of macrophages in Ctrl, APS (APP in the revised manuscript) and APR tumors, we conducted enrichment analysis by ssGSEA by using literature-based M1/M2 genesets as listed below. As shown in **Fig.R9**, surprisingly, no significant difference of M1/M2 phenotypes of macrophages was found.

M1/M2 gene sets

M1 genes	Ccl2	Ccl3	Ccl5	Ccl8	Ccl9	Ccl10	Ccl11	Ccl15	Ccl19	Cxcl9	Cxcl10	Cxcl11	Cd86	Cd80	Il12a	Il1b	Il6	Irf5	Nos2	Ptgs2	Socs3	Tnf
M2 genes	Arg1	Ccl17	Ccl22	Ccl24	Chil3	Chil4	Cxcr2	Egr2	Fabp4	Fn1	Il10	Il1ra	Irf4	Klf4	Mrc1	Pparg	Retnla	Tgfb1				

Fig.R9 Examination of M1/M2 phenotypes of macrophages by ssGSEA (for reviewers only).

3. The set-up of experiment shown in Fig 3G is not entirely clear. Were CD45+ cells first enriched from the tumor and then Gr1+ cells enriched from the CD45+ fraction? Otherwise, we would expect Gr1+ cells to be a very big part of the CD45+ population, especially because APR tumors were analyzed for this experiment (around 80% based on the authors previous scRNAseq in Fig 2C). Similarly, group I cells would not show as high expression of macrophage and T/NK cell genes in the total CD45+ population.

Reply: Thank you for asking for clarification. We believe the reviewer's comment is referring to the original **Fig. 2G**, instead of **Fig. 3G**. Our experiment set-up is indeed as the reviewer understood. The dissociated tumor suspension was firstly divided into two parts, then one part for CD45+ TILs MACS and the other for Gr-1+ cells MACS side-by-side. We apologize for the lack of clarity on the sources of CD45+ TILs and Gr-1+ cells. The tumor samples shown in **Fig 2G** were derived from the APR tumor transplantation model. We have now specified this in our revised manuscript (Please see Page 8, line 4-7 of 2nd paragraph, blue text). Although the percentage of Gr-1+ cells appeared to vary between resistant tumors from the spontaneous and transplanted model, in line with the reviewer's expectation and interpretation, a significant part (~40%) of the CD45+ population in **Fig. 2H** was clustered into group II cells.

4. Confirm activation of T cell anti-tumor activity observed in scRNAseq in Figure 4G with other methods, such as IHC of granzyme B.

Reply: Thank you for this suggestion. We have added the result of granzyme B expression in CD8+ T cells, as shown in **Fig.R10** (Please see **Supplementary Fig. 10F** in the revised manuscript).

Fig.R10 Granzyme B expression in CD8+ T cells by IF staining.

Minor points:

1. The authors used Ab+Pal treatment to obtain resistant tumors. In the text and several figures (e.g. Fig 1B, Fig 1H, Fig 2K), the authors refer to the continued treatment as “days post treatment”. This should probably be “days of treatment” or similar, as treatment was continued and not stopped.

Reply: We thank the reviewer for the correction. We have replaced “days post treatment” with “days of treatment” in our revised manuscript.

2. It would be more better if the authors included actual p-values instead of asterisks in all their figures.

3. Include p-values for Figures 2K, 3D, 5F, S7A. Include p-values for CyTOF data (Figures 2E, 4A).

Reply: As suggested by the reviewer, we have now included P-values in the related figures in our revised manuscript. To better show the p-values of the CyTOF results, we have split CD11b^{hi} Ly6C/G^{hi}, CD11b^{hi} Ly6C/G^{low} Cx3cr1^{low} and CD8 cells and re-plotted them as shown in **Fig.R11** (new **Fig. 2D** and **Fig. 4A**) with error bars.

Fig.R11 Re-plotted CyTOF results (with P-values and error bars).

4. Similarly, it would be useful if the authors included Hazard ratios of their Kaplan-Meier curves in Figures S1B, 3F, 5B.

Reply: Thank you for this suggestion. In the original Fig.3F (Fig.3I in the revised manuscript), the computed hazard ratio by logrank approach for Ab+Cabo vs Ctrl is 0.2268, and for Ab+Cabo+ICB vs Ctrl is 0.2146. Given that the population hazard ratio in our case is not proportional and constant over the study time in different treatment groups. For example, in Fig.3F, all mice in Ctrl group have reached the endpoint in 7 days, while none in Ab+Cabo and Ab+Cabo+ICB groups have reached endpoint at day 7, which could be problematic for interpretation the hazard ratio at day 7 for Ab+Cabo and Ab+Cabo+ICB groups. In addition, the hazard ratio does not provide direct survival/time-to-event information. We feel that hazard ratio might be an over interpretation while less meaningful in the situations of Figure.S1B,3I and 5B.

5. *Figure legends should include number of animals/group used (Figures S1B, 3F, 5B)*

Reply: We thank the reviewer for the suggestion. We have included the number of animals used in each group in the revised Figure legends.

6. *Spelling in Figures 1B & S1C: Volume instead of Volumn*

Reply: We thank the reviewer for the correction. We have corrected this in our revised manuscript.

7. *Gene names should be italicized, such as in Figures S2D&H, S3B-E, S5C*

Reply: We thank the reviewer for the correction. Those gene names have been italicized in our revised manuscript.

8. *Check colors in Figure S3C, figure and legend do not correspond*

Reply: Thank you for pointing out this. We have solved this issue in the revised manuscript.

9. *It would be helpful for the reader if the authors described in a bit more detail the markers used particularly in the CyTOF experiment in Fig 2E. They do so partially later in the text, but it's recommended that any marker mentioned for the first time is explained briefly as well as the rationale for using this marker. Graphs with markers should have an explanation what the marker is for, so readers not familiar with immunology can quickly understand the context.*

Reply: Following the reviewer's suggestion, we have added a brief explanation regarding Ly6C/G (Gr1) and Cx3cr1 in Page 7 in the revised manuscript main text.

10. *It would be helpful for the reader if t-SNE plots in the main figures would include the group numbers, so the reader does not have to look those up in the supplemental figures.*

Reply: Thank you for this suggestion. We have added the group numbers onto t-SNE plots in **Fig. 2H**.

REVIEWERS' COMMENTS:

Reviewer #1 (Remarks to the Author):

The authors have done a good job of addressing the reviewer's concerns, and the manuscript is significantly improved. No additional comments-

Reviewer #2 (Remarks to the Author):

The authors have made the necessary changes to address my concerns.

Reviewer #3 (Remarks to the Author):

The authors have addressed each of the specific points raised by the reviewers and revised the manuscript accordingly. The revised manuscript is significantly improved.

Re: Final revisions for manuscript NCOMMS-19-00732A

REVIEWERS' COMMENTS:

Reviewer #1 (Remarks to the Author):

The authors have done a good job of addressing the reviewer's concerns, and the manuscript is significantly improved. No additional comments-

Reviewer #2 (Remarks to the Author):

The authors have made the necessary changes to address my concerns.

Reviewer #3 (Remarks to the Author):

The authors have addressed each of the specific points raised by the reviewers and revised the manuscript accordingly. The revised manuscript is significantly improved.

Reply:

We again thank the editor and the reviewers for the time and contributions to our manuscript.